# Exploring beyond clinical routine SARS-CoV-2 serology using MultiCoV-Ab to evaluate endemic coronavirus cross-reactivity

Matthias Becker [1,24], Monika Strengert[2,3,24], Daniel Junker[1], Philipp D. Kaiser[1], Tobias Kerrinnes[4], Bjoern Traenkle[1,5], Heiko Dinter[1,5], Julia Häring[1], Stéphane Ghozzi[2], Anne Zeck[1], Frank Weise[1], Andreas Peter[6,7,8], Sebastian Hörber [6,7,8], Simon Fink [1], Felix Ruoff[1], Alex Dulovic [1], Tamam Bakchoul [9], Armin Baillot[10], Stefan Lohse [11], Markus Cornberg[12], Thomas Illig[13], Jens Gottlieb[14,15], Sigrun Smola[11], André Karch[16], Klaus Berger[16], Hans-Georg Rammensee[17,18,19], Katja Schenke-Layland [1,19,20,21], Annika Nelde [17,19,22], Melanie Märklin [19,21], Jonas S. Heitmann[19,21], Juliane S. Walz [17,19,23,22], Markus Templin[1], Thomas O. Joos[1], Ulrich Rothbauer[1,5,24], Gérard Krause [2,3] & Nicole Schneiderhan-Marra [1✉]

The humoral immune response to SARS-CoV-2 is a benchmark for immunity and detailed analysis is required to understand the manifestation and progression of COVID-19, monitor seroconversion within the general population, and support vaccine development. The majority of currently available commercial serological assays only quantify the SARS-CoV-2 antibody response against individual antigens, limiting our understanding of the immune response. To overcome this, we have developed a multiplex immunoassay (MultiCoV-Ab) including spike and nucleocapsid proteins of SARS-CoV-2 and the endemic human coronaviruses. Compared to three broadly used commercial in vitro diagnostic tests, our MultiCoV-Ab achieves a higher sensitivity and specificity when analyzing a well-characterized sample set of SARS-CoV-2 infected and uninfected individuals. We find a high response against endemic coronaviruses in our sample set, but no consistent cross-reactive IgG response patterns against SARS-CoV-2. Here we show a robust, high-content-enabled, antigen-saving multiplex assay suited to both monitoring vaccination studies and facilitating epidemiologic screenings for humoral immunity towards pandemic and endemic coronaviruses.

A full list of author affiliations appears at the end of the paper.

Since its first characterization in late 2019, SARS-CoV-2, the seventh known coronavirus to infect humans, has developed into a worldwide pandemic with dramatic socio-economic consequences[1–3]. While the majority of individuals suffer only from mild symptoms, approximately 14% of infected adults experience particularly severe disease outcomes (i.e., pneumonia) of COVID-19[4]. Of these 14%, 5% will progress into a critical condition characterized by hypoxaemic respiratory failure, acute respiratory distress syndrome and multiorgan failure[4]. To date, the large number of infected individuals and of those requiring urgent intensive care has put a high burden on public healthcare infrastructures[3].

In contrast to the recently emerged SARS-CoV-2, the four endemic human coronaviruses (hCoVs) NL63 and 229E (α-hCoVs) as well as OC43 and HKU1 (β-hCoVs), regularly circulate in the population[5] (one study reported a 91% prevalence for OC43 in the adult population[6]) and are thought to cause up to 20% of mild colds[7]. As humoral immune responses are in general seen as protective by production of neutralizing antibodies to viral surface proteins[8], it would be tempting to speculate that a previous infection with an endemic strain offers protection against infection with the β-coronavirus SARS-CoV-2, as already seen in in vitro studies[9]. However, it has also been reported that for both SARS coronaviruses and MERS-CoV, disease severity and fatal outcome correlates with early seroconversion and/or increased antibody titers by a yet undefined mechanism[10–13]. Consequentially, a detailed understanding of the humoral SARS-CoV-2 immune response is of importance to provide insights into COVID-19 disease biology[12,14].

Serological tests are essential tools in cohort-based epidemiological studies to determine seroprevalence and precisely assess mortality rates, the extent of asymptomatic or mild infections not currently detectable by molecular testing, and ultimately determine the effectiveness of population-based interventions and direct future preventive strategies. Furthermore, serological testing is a companion diagnostic to monitor vaccination efficacy and mode of action in vaccine trials[15,16]. As a result, there is a need for robust serological tests to quantify antibody production against SARS-CoV-2 in detail. Currently, most commercially available serological assays utilize single analyte technologies (i.e., ELISA) to measure antibodies against SARS-CoV-2 spike (S) or nucleocapsid (N) antigens[16–19]. Few tests combine and correlate N- and S-antigen-based detection[20–22] or attempt global profiling of antibody responses against the entire SARS-CoV-2 genome[23]. To this end, we developed a multiplexed SARS-CoV-2 immunoassay (MultiCoV-Ab) which included not only S and N protein-based antigens of SARS-CoV-2, but also from endemic hCoVs (NL63, 229E, OC43, HKU1) based on findings of numerous SARS-CoV-1 serological studies, which reported on cross-reactive antibodies to antigens from circulating hCoVs[24]. Such an expanded antigen panel allows to both resolve the SARS-CoV-2 antibody response in detail and to assess and correlate potential cross-protection mechanisms between coronaviruses. We measured both IgA and IgG responses, as these isotypes in contrast to IgM can persist for extended periods in the serum and in nasal fluids[25]. Further, SARS-CoV-2 is a mucosal-targeted virus, and reports indicate that IgA, as the dominant antibody isotype in the mucosal defense is a good indicator for early immune defense mechanisms in this case[26].

In this study, to determine how well MultiCoV-Ab performs, we compare our assay to broadly applied commercial in vitro diagnostic (IVD) tests with well-characterized sample sets for clinical validation and further analyze potential sources of cross-reactivity with hCoVs. For the sample set examined, we were able to reach a specificity of 100% with MultiCoV-Ab and achieved an improved sensitivity compared to commercial tests, confirming its value as a serological screening assay.

## Results

**MultiCoV-Ab: a highly sensitive test for SARS-CoV-2 seroconversion**. To investigate the antibody response of SARS-CoV-2-infected individuals, we developed and established a high-throughput and automatable bead-based multiplex assay, termed MultiCoV-Ab. We expressed and immobilized six different SARS-CoV-2-specific antigens on Luminex MAGPLEX beads with distinct color codes, specifically the trimeric full-length spike protein (Spike Trimer), receptor-binding domain (RBD), S1 domain (S1), S2 domain (S2), full-length nucleocapsid (N), and the N-terminal domain of nucleocapsid (N-NTD) (Supplementary Fig. 1). Immunoglobulins from serum and plasma samples were detected using phycoerythrin-labeled anti-human IgG or IgA antibodies. To ensure assay stability and comparability, quality control samples were processed in parallel within every assay run. Quality control and assay performance data sets are provided in Supplementary Fig. 2 and Supplementary Table 1.

To analyze SARS-CoV-2-induced seroconversion, we used the Spike Trimer and RBD (previously described by Amanat et al.[27]) as key antigens for classification, and initially screened a sample set of 205 reconvalescent SARS-CoV-2-infected and 72 uninfected individuals with the MultiCoV-Ab. To critically assess assay performance, we compared our results with three commercially available IVD tests widely used in clinical routine SARS-CoV-2 antibody testing namely: Elecsys Anti-SARS-CoV-2 (antibodies including IgG; Roche[28]), SARS-CoV-2 Total (total antibodies IgM and IgG; Siemens Healthineers[29]) and Anti-SARS-CoV-2 ELISA (IgG/IgA; Euroimmun[30]). Using a combined cut-off of both antigens, we identified all uninfected samples as negative (Fig. 1a). In accordance with our MultiCoV-Ab, none of the uninfected samples was classified as false positive by the Roche and Siemens tests, while one sample was classified as false positive and one as "borderline" by the Euroimmun IgG test. Of the 205 infected samples, both MultiCoV-Ab and commercial IVD tests for total Ig or IgG identified 24 (11.7%) as IgG antibody-negative. However, the IVD tests missed an additional 8 (Roche), 11 (Siemens Healthineers), and 9 (Euroimmun IgG) samples of SARS-CoV-2-infected individuals. Furthermore, the Euroimmun IgG test classified 8 additional samples as "borderline" (Fig. 1b, Supplementary Fig. 3a–c). When testing for IgA antibodies in serum/plasma of SARS-CoV-2-infected individuals, our MultiCoV-Ab classified 47 (22.9%) as IgA-negative, whereas the Euroimmun test classified 32 (15.6%) as IgA-negative, and 16 (7.8%) as borderline (Fig. 1b and Supplementary Fig. 3d). For the uninfected samples, the Euroimmun IgA test identified 7 (9.7%) as false positives and 3 (4.2%) as "borderline", whereas no samples were classified as false positives by the MultiCoV-Ab. Overall, the MultiCoV-Ab achieved a sensitivity of 88.3% and a specificity of 100% in this initial set of samples using IgG detection (Table 1). When comparing the results of the commercial IVD tests to the respective manufacturers' specifications, all tests were unable to reach their stated sensitivity of 100%. In contrast, for all commercial tests, the found specificities were close to the manufacturers' stated specificity in our sample set (Table 1). This demonstrates that antigen selection and assay setup are crucial in achieving optimal performance and must be considered when screening for SARS-CoV-2, particularly in low prevalence scenarios.

**Multiplex serology improves assay specificity**. Next, to perform a more detailed clinical validation of our MultiCoV-Ab, we expanded our sample set to a total of 310 SARS-CoV-2-infected and 866 uninfected donors (a simplified overview of this set is shown in Table 2; a complete breakdown is displayed in Supplementary Table 2). We performed a ROC analysis[31,32] per

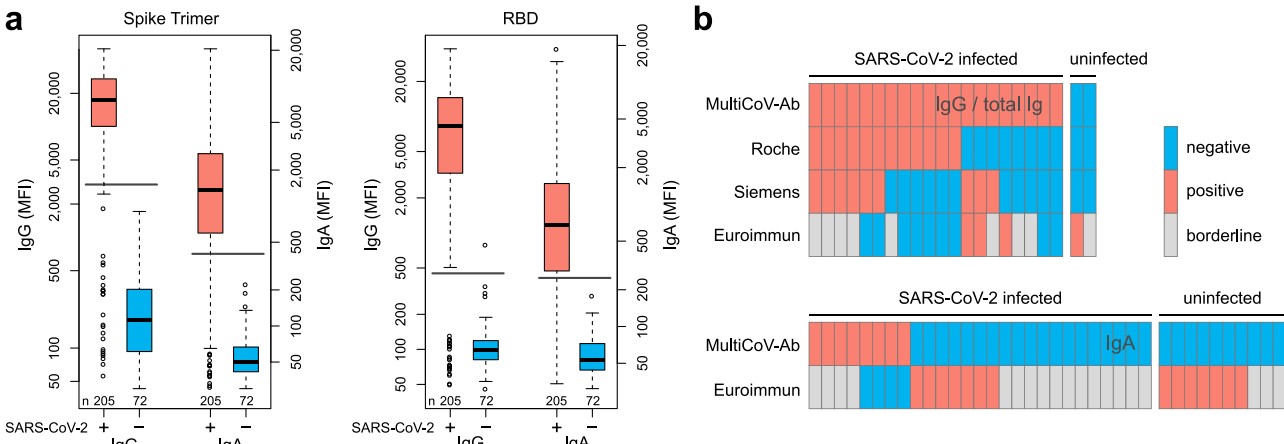

**Fig. 1 MultiCoV-Ab, a sensitive and specific tool to monitor SARS-CoV-2 antibody responses. a** Control sera (blue, $n = 72$) and sera from individuals with PCR-confirmed SARS-CoV-2 infection (red, $n = 205$) were screened in a multiplex bead-based assay using Luminex technology (MultiCoV-Ab) to quantify IgG or IgA responses to various antigens. Reactivity towards trimeric SARS-CoV-2 spike protein (Spike Trimer) or SARS-CoV-2 receptor binding domain of spike (RBD) was found to be the best predictor of SARS-CoV-2 infection. Data are presented as Box-Whisker plots of a sample's median fluorescence intensity (MFI) on a logarithmic scale. Box represents the median and the 25th and 75th percentiles, whiskers show the largest and smallest values. Outliers determined by 1.5 times IQR of log-transformed data are depicted as circles. Cut-off values for classification for single antigens are displayed as horizontal lines (Spike Trimer IgG: 3,000 MFI, IgA: 400 MFI; RBD IgG: 450 MFI, IgA: 250 MFI). **b** Sample set from **a**, was used to compare assay performance of the MultiCoV-Ab using Spike Trimer and RBD antigens with commercially available single analyte SARS-CoV-2 IVD assays which detect total Ig (Elecsys Anti-SARS-CoV-2 (Roche); ADVIA Centaur SARS-CoV-2 Total (COV2T) (Siemens Healthineers)) or IgG (Anti-SARS-CoV-2-ELISA - IgG (Euroimmun)) or IgA (Anti-SARS-CoV-2-ELISA - IgA (Euroimmun)). SARS-CoV-2 infection status of samples based on PCR diagnostic is indicated as SARS-CoV-2 positive or negative. Antibody test results were classified as negative (blue), positive (red), or borderline (gray) as per the manufacturer's definition. Only samples with divergent antibody test results are shown. **c** Performance and specifications as stated in the manufacturer's IVD assay manual. For the manufacturer sensitivity specification, information for samples >14 days post-infection are presented. Respective sensitivity and specificity values calculated in this study are given with 95% Clopper-Pearson confidence intervals[52]. Positive and negative predictive values (PPV/NPV) were calculated based on a seropositivity of 3%. Source data are provided as a Source Data file.

SARS-CoV-2 antigen and detection system (Supplementary Fig. 4), which confirmed that Spike Trimer and RBD were the best predictors of SARS-CoV-2 infection. We, therefore, decided to use a combination of both antigens (IgG or IgA overall cut-off) to define overall SARS-CoV-2 reactivity for IgG or IgA, for which the two independent cut-offs for Spike Trimer and RBD had to be met (Table 3). Cut-offs were chosen with focus on maximum specificity for the overall classification (Spike Trimer+/RBD+) to prevent false positive results (Fig. 2a). With the overall IgG cut-off, we reached a specificity of 100%, which would not have been possible for either of the antigens individually, while still retaining acceptable sensitivity (88.7%). IgG detection was shown to be more specific and sensitive than IgA for determination of SARS-CoV-2 infection within our sample set. Only 8 samples which were IgA-positive showed no IgG response (Fig. 2b, dashed lines), 2 of which were uninfected and falsely classified as positive. Of the 6 remaining samples, metadata (including the time between the onset of symptoms and sample collection) was available for 4 (2, 6, 7, and 15 days). As a result, we hypothesized that IgA in these samples can be used to measure an early onset of antibody response as has been proposed by several groups[26,33,34]. Therefore, to give an overall measure of SARS-CoV-2 infection, we used the IgG classification as a basis and included samples with strong IgA positivity–signal to cut-off (S/CO) > 2 for Spike Trimer and RBD–as positive, irrespective of their detected IgG response (Fig. 2b, straight lines). With this combined IgG + IgA classification, we reached an optimal sensitivity of 90% while retaining a specificity of 100%.

**Antigen selection affects SARS-CoV-2 serology test performance.** While further analyzing the immune response detected towards our 4 additional SARS-CoV-2 antigens in our Multiplex panel, we assessed the IgG response towards the S1 and S2 subdomains of the spike, which both did not improve sample classification (Fig. 2c). Interestingly, RBD, which is a part of S1, showed fewer uninfected samples with increased IgG response compared to S1. For S2, even more, uninfected samples had increased signals, suggesting the presence of potential cross-reactive antibodies for this domain of the spike protein (Fig. 2c). These findings suggest that the RBD response is highly characteristic of the overall SARS-CoV-2 immune response. To further complement our assay, we included the N and N-NTD proteins. Although these antigens have been successfully used in single-analyte assays[35], we observed a high cross-reactivity in uninfected samples for both (Fig. 2d). Interestingly, across the entire data set, only one sample showed a distinct immune response to N and N-NTD, but not to all spike-derived antigens. This confirms that the performance of an antigen is specific to the assay setup and cannot be easily generalized, as commercial IVD tests (i.e., Roche) are able to use the N protein to great effect in a different assay setup.

**Dynamics of antibody response in COVID-19 patients.** Longitudinal samples from 5 hospitalized patients were used to perform a small-scale time-course analysis of IgG and IgA immune responses (Fig. 3a). Levels of both Ig classes strongly increased within the first ten days after the onset of symptoms. While IgG levels appeared constant over roughly two months, IgA levels started to decline between day 10 and 20 after the onset of symptoms. This reduction in IgA antibody levels was also observed with increased time post-infection in samples without longitudinal follow-up (Supplementary Figure 5). These effects were consistent for the majority of SARS-CoV-2 antigens.

**Table 1 Comparison of MultiCoV-Ab and commercial IVD tests in screening results and manufacturer specifications as stated in the assay manuals.**

| Assay | Detection | Antigen used | Manufacturer specification >14 days post infection | | Correctly classified | | Found sensitivity | Found specificity | PPV at 3% | NPV at 3% |
|---|---|---|---|---|---|---|---|---|---|---|
| | | | Sensitivity | Specificity | Infected (of 205) | Uninfected (of 72) | (95% CI) | (95% CI) | Prevalence | Prevalence |
| MultiCoV-Ab | IgG | S Trimer + RBD | – | – | 181 | 72 | 88.3% (83.1–92.4%) | 100% (95.0–100%) | 100% | 99.6% |
| Roche | Total Ig | N | 100% n = 29 | 99.8% n = 5272 | 173 | 72 | 84.4% (78.7–89.1%) | 100% (95.0–100%) | 100% | 99.5% |
| Siemens | Total Ig | S1 RBD | 100% n = 47 | 99.8% n = 1589 | 170 | 72 | 82.9% (77.1–87.8%) | 100% (95.0–100%) | 100% | 99.5% |
| Euroimmun | IgG | S1 | 100% n = 13 | 99.0% n = 1261 | 164 | 70 | 80.0% (73.9–85.2%) | 97.2% (90.3–99.7%) | 46.9% | 99.4% |
| MultiCoV-Ab | IgA | S Trimer + RBD | – | – | 158 | 72 | 77.1% (70.7–82.6%) | 100% (95.0–100%) | 100% | 99.3% |
| Euroimmun | IgA | S1 | 100% n = 13 | 90.4% n = 1261 | 157 | 62 | 76.6% (70.2–82.2%) | 86.1% (75.9–93.1%) | 14.6% | 99.2% |

**Table 2 Extended Sample set used to further validate MultiCoV-Ab performance.**

| Age group | ≤39 | | 40-59 | | ≥60 | | Not available | | Σ |
|---|---|---|---|---|---|---|---|---|---|
| n | 299 (25.4%) | | 241 (20.5%) | | 475 (40.4%) | | 161 (13.7%) | | 1176 |
| Sex | Male | Female | Male | Female | Male | Female | Male | Female | |
| n | 139 (11.8%) | 160 (13.6%) | 144 (12.2%) | 97 (8.2%) | 271 (23.0%) | 204 (17.3%) | 5 (0.4%) | 3 (0.3%) | 1176 |
| SARS-CoV-2 infected | 60 (19.4%) | 51 (16.5%) | 71 (22.9%) | 63 (20.3%) | 42 (13.5%) | 17 (5.5%) | 3 (1.0%) | 3 (1.0%) | 310 |
| hospitalized | 6 (10.9%) | 2 (3.6%) | 14 (25.5%) | 6 (10.9%) | 23 (41.8%) | 4 (7.3%) | 0 (0.0%) | 0 (0.0%) | 55 |
| non-hospitalized | 52 (25.0%) | 43 (20.7%) | 49 (23.6%) | 43 (20.7%) | 13 (6.3%) | 8 (3.8%) | 0 (0.0%) | 0 (0.0%) | 208 |
| hospitalization NA | 2 (4.3%) | 6 (12.8%) | 8 (17.0%) | 14 (29.8%) | 6 (12.8%) | 5 (10.6%) | 3 (6.4%) | 3 (6.4%) | 47 |
| SARS-CoV-2 uninfected | 79 (9.1%) | 109 (12.6%) | 73 (8.4%) | 34 (3.9%) | 229 (26.4%) | 187 (21.6%) | 2 (0.2%) | 0 (0.0%) | 866 |

**Table 3 MultiCoV-Ab sensitivity and specificity of extended sample set for IgA and IgG based on a single analyte or a combined cut-off of Spike Trimer and RBD (IgG or IgA overall) or combined isotype cut-off (IgG and IgA).**

| | Correctly classified | | Sensitivity | Specificity | PPV at 3% | NPV at 3% |
|---|---|---|---|---|---|---|
| | Infected | Uninfected | (95% CI) | (95% CI) | Prevalence | Prevalence |
| IgG Spike Trimer | 277 | 849 | 89.4% (85.4–92.6%) | 98.0% (96.9–98.9%) | 58.5% | 99.7% |
| IgG RBD | 276 | 862 | 89.0% (85–92.3%) | 99.5% (98.8–99.9%) | 85.7% | 99.7% |
| IgG overall | 275 | 866 | 88.7% (84.6–92%) | 100% (99.6–100%) | 100% | 99.7% |
| IgA Spike Trimer | 272 | 850 | 87.7% (83.6–91.2%) | 98.2% (97–98.9%) | 59.5% | 99.6% |
| IgA RBD | 255 | 855 | 82.3% (77.5–86.3%) | 98.7% (97.7–99.4%) | 66.7% | 99.4% |
| IgA overall | 254 | 864 | 81.9% (77.2–86.1%) | 99.8% (99.2–100%) | 91.7% | 99.4% |
| Combined IgA & IgG | 279 | 866 | 90.0% (86.1–93.1%) | 100% (99.6–100%) | 100% | 99.7% |

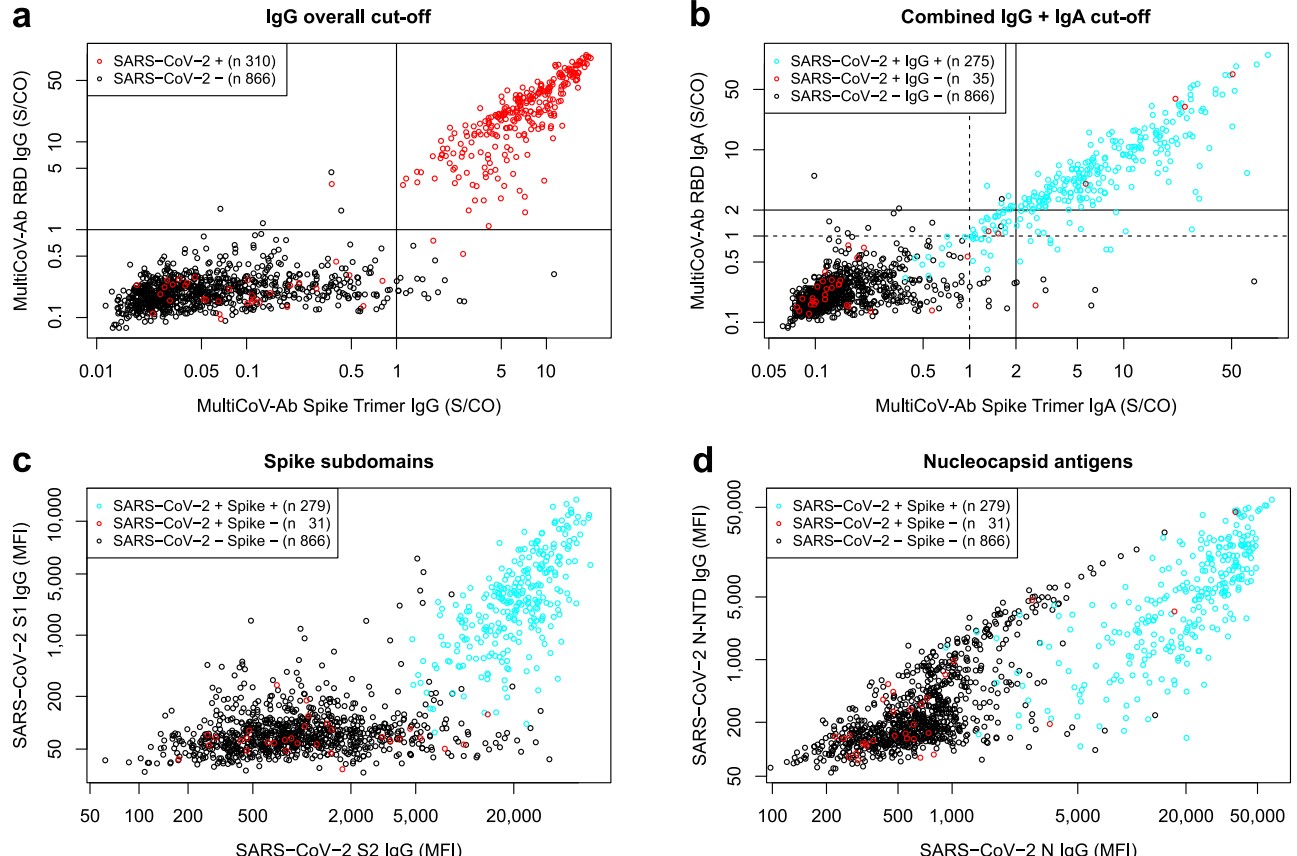

**Fig. 2 Combination of 2 spike protein variants and isotype profiling by multiplex assay increases accuracy to identify SARS-CoV-2 antibody-positive individuals. a, b** Scatterplot detailing MultiCoV-Ab cut-offs. Signal to cut-off (S/CO) values are displayed for Spike Trimer against RBD on a logarithmic scale. For IgG (**a**), cut-offs are visualized by straight lines and SARS-CoV-2-infected and uninfected samples are separated by color (black circles – SARS-CoV-2-uninfected; red circles – SARS-CoV-2-infected). For IgA (**b**) cut-offs are visualized as dashed lines and S/CO of 2 used for the combined cut-off is shown as straight lines. SARS-CoV-2-infected samples are split into IgG-positives and -negatives by color as indicated in the plot. **c, d** Scatterplots display IgG response to additional SARS-CoV-2 antigens contained in the MultiCoV-Ab panel: MFI for spike subdomains S1 vs S2 (**c**) or nucleocapsid antigens N vs N-NTD (**d**) are displayed on a logarithmic scale. SARS-CoV-2-uninfected samples are distinguished from SARS-CoV-2-infected and MultiCoV-Ab classification into positives or negatives as indicated by color. Source data are provided as a Source Data file.

Furthermore, we found that patients' hospitalization, as a measure of disease severity (Fig. 3b), seemed to correlate with an increased humoral immune response, particularly for IgA. Lastly, we identified a trend for increasing age (Fig. 3c). While we overall see correlation of the immune response with patient hospitalization, age, and time post-infection, our sample set was not designed to single out the leading cause amongst these effects. Patients of higher age also had a higher rate of hospitalization in our study population (see Table 2) and samples with increased time post-infection were also less often hospitalized. It should also be noted that our samples from infected donors had different origins and thus different determinants of time post-infection, as some were based upon PCR results and others on symptom onset.

**Previous endemic hCoV infection indicates higher immune response to SARS-CoV-2.** In order to explore cross-reactivity of hCoVs with SARS-CoV-2, we included S1, N, and N-NTD

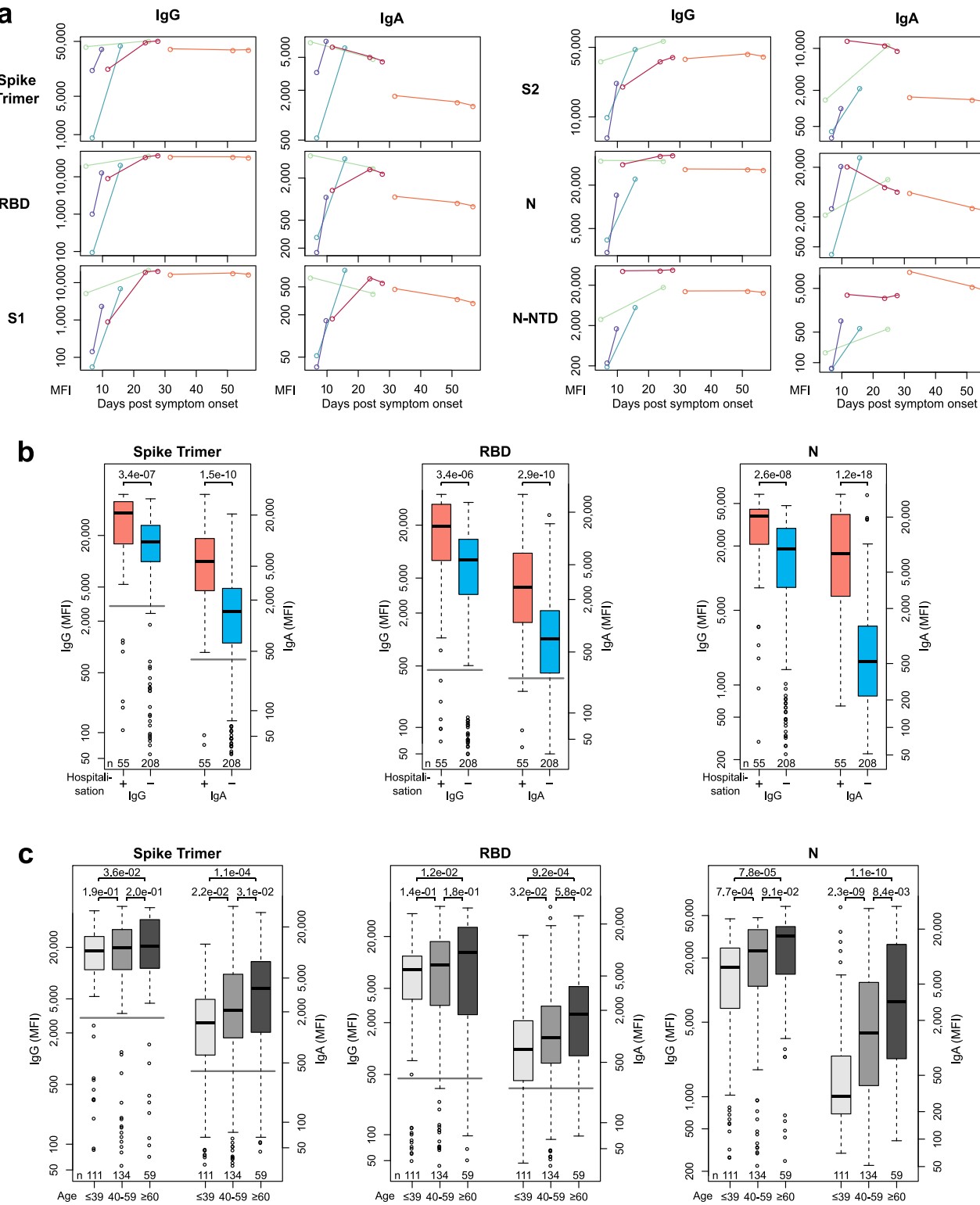

**Fig. 3 Multiplex-based seroprofiling allows in-depth characterization of SARS-CoV-2 antibody responses. a** Kinetic of SARS-CoV-2 antigen-specific IgA and IgG responses is shown for indicated days after symptom onset for six SARS-CoV-2-specific antigens for five different patients. Patients are indicated by color. **b**, **c** Samples of SARS-CoV-2-infected individuals were analyzed to identify antigen- and isotype-specific antibody responses based on hospitalization indicating disease severity (**b**) or age (**c**). Data is presented as Box-Whisker plots of sample MFI on a logarithmic scale. Box represents the median and the 25th and 75th percentiles, whiskers show the largest and smallest values. Outliers determined by 1.5 times IQR of log-transformed data are depicted as circles. *p*-value (Mann–Whitney U test, two-sided) is displayed at the top of the boxes, indicating differences between signal distribution for respective groups. Cut-off values for MultiCoV-Ab classification are displayed as horizontal lines (Spike Trimer IgG: 3,000 MFI, IgA: 400 MFI; RBD IgG: 450 MFI, IgA: 250 MFI). Source data are provided as a Source Data file.

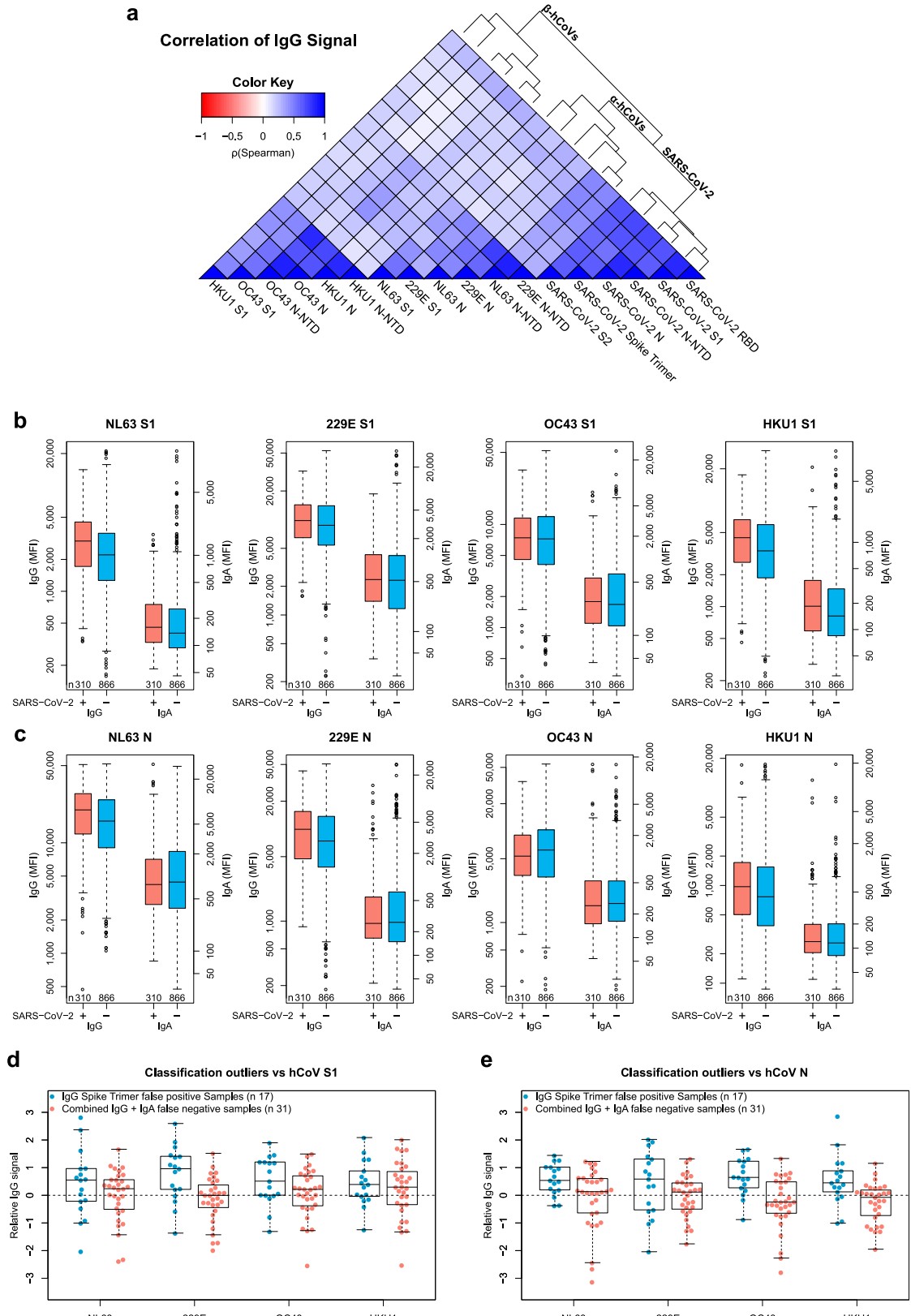

antigens from human α- (NL63 and 229E) and β-hCoVs (OC43 and HKU1) in our MultiCoV-Ab panel (Supplementary Fig. 1). The immune response towards all hCoV antigens was more dependent on coronavirus clade than on antigen choice. However, within the clades of α-hCoVs and β-hCoVs, types of antigens were more dominant than the virus subtype, as

demonstrated by rank correlation analysis and hierarchical clustering (Fig. 4a, Supplementary Fig. 6a), suggesting there is potential cross-reactivity within the hCoV clades. Interestingly, IgG response against α-hCoVs clustered more closely to SARS-CoV-2 than to β-hCoVs. This is unexpected, since SARS-CoV-2 has been assigned to the clade of β-CoVs and is also more similar

**Fig. 4 Correlation of seasonal hCoV and SARS CoV-2 antibody responses. a** Correlation of IgG response for the entire sample set ($n = 1176$) is visualized as heatmap based on Spearman's $\rho$ coefficient; dendrogram on the right side displays antigens after hierarchical clustering was performed. **b-c,** Immune responses (IgG and IgA) towards hCoV S1 (**b**) and N (**c**) proteins are presented as Box-Whisker plots of sample MFI on a logarithmic scale for SARS-CoV-2-infected (red, $n = 310$) and uninfected (blue, $n = 866$) individuals. Box represents the median and the 25th and 75th percentiles, whiskers show the largest and smallest values. Outliers determined by 1.5 times IQR of log-transformed data are depicted as circles. **d-e,** Relative levels of IgG-specific immune response towards hCoV S1 (**d**) and N (**e**) proteins are presented as Box-Whisker plots/strip chart overlays of log-transformed and per-antigen scaled and centered MFI for the sample subsets of Spike Trimer false positives (blue, $n = 17$) and combined IgG + IgA false negatives (red, $n = 31$). Box represents the median and the 25th and 75th percentiles, whiskers show the largest and smallest values, excluding outliers as determined by 1.5 times IQR. Source data are provided as a Source Data file.

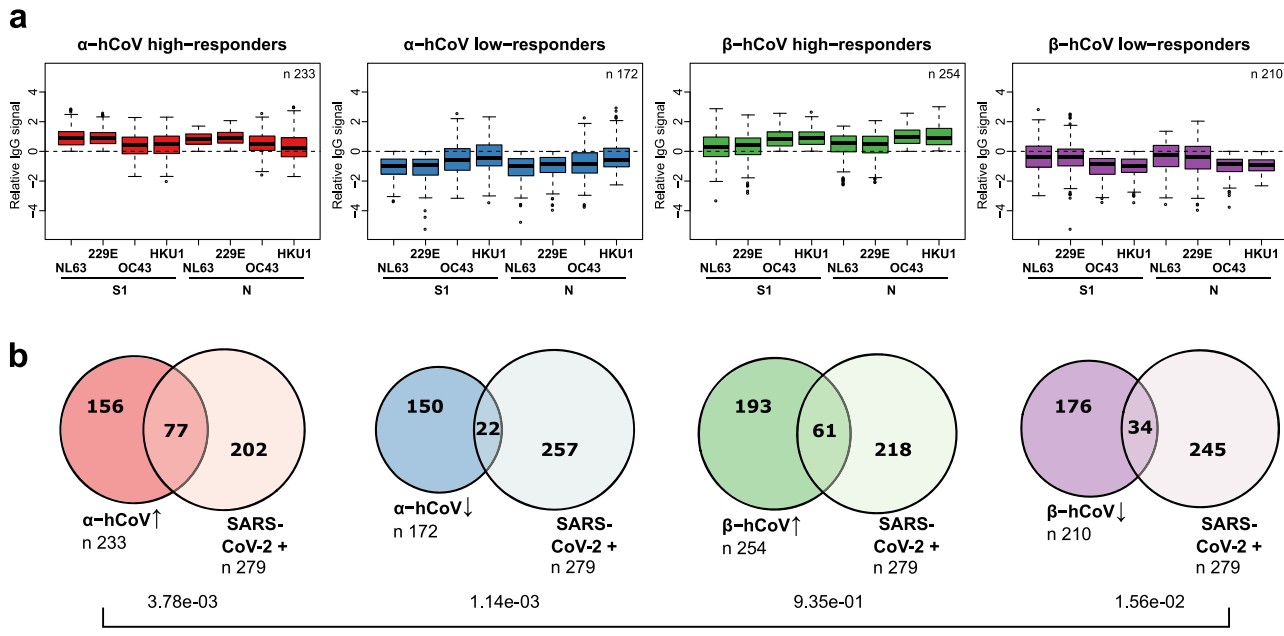

**Fig. 5 Analysis of seasonal hCoV high and low responders. a** From the entire study population, groups of α- or β-hCoV high and low responders were built as indicated. High responder were defined as samples with above average MFI values for S1 and N-specific IgGs of the respective hCoV clade. Low responders were defined with below MFI values, correspondingly. Responder groups (i) α-hCoV ↑, red, $n = 233$, (ii) β-hCoV ↑, green, $n = 254$, (iii) α-hCoV ↓, blue, $n = 172$ (iv) β-hCoV ↓, purple, $n = 210$ are shown as Box-Whisker plots of log-transformed and per-antigen scaled and centered MFI values across hCoV N and S1 antigens. Box represents the median and the 25th and 75th percentiles, whiskers show the largest and smallest values. Outliers determined by 1.5 times IQR are depicted as circles. **b** The over- or under-representation of SARS-CoV-2 responders (SARS-CoV-2 + , $n = 279$, as determined by positive MultiCoV-Ab classification) within the four sample groups is visualized in Venn diagrams, stochastic significance was calculated using Fisher's exact test (two-sided). Source data are provided as a Source Data file.

in sequence to the β-hCoVs (Supplementary Table 3). Overall, we identified a considerable immune response to hCoV antigens throughout the whole sample set with no notable differences between samples from SARS-CoV-2-infected and uninfected donors in IgG or IgA for S1 (Fig. 4b), N (Fig. 4c), or N-NTD (Supplementary Fig. 6b). We, therefore, used the IgG signal relative to the average response per antigen for further analyses, which allowed comparison among all hCoV antigens on one scale. For those uninfected samples which showed an IgG cross-reactivity towards Spike Trimer (Spike Trimer false positives), we partially observed increased responses towards hCoV antigens. Those samples, which did not show an immune response after SARS-CoV-2 infection (false negatives, as determined by Multi-CoV-Ab, combined IgG + IgA) were closer to the baseline (Fig. 4d, e, Supplementary Fig. 6c). This indicates that cross-reactivity with hCoVs causes some of the observed SARS-CoV-2 immune response in samples taken from individuals not exposed to SARS-CoV-2. To investigate the correlation of hCoV and SARS-CoV-2 immune response further, we grouped samples into high and low responders for α-hCoVs and β-hCoVs, as the

antigens were shown to correlate closely within a single hCoV clade. High responders were defined as having relative IgG signals > 0 for N and S1 antigens of both hCoV subtypes within the clade, while low-responders had signals < 0, respectively (Fig. 5a). Samples with SARS-CoV-2 immune response (as determined by MultiCoV-Ab, combined IgG + IgA classification) were significantly overrepresented within the group of α-hCoV high responders ($p = 3.78e{-}03$, Fisher's exact test, two-sided), while being significantly underrepresented within the group of α-hCoV and β-hCoV low responders ($p = 1.14e{-}03$ and $p = 1.56e{-}02$, respectively, Fisher's exact test, two-sided) (Fig. 5b). These results showed that while there were no discernible global effects for single antigens, there is a correlation between the SARS-CoV-2 immune response with high hCoV responses, especially towards α-hCoVs. This effect, and the clustering of α-hCoVs and SARS-CoV-2 in Fig. 4a may be a result of similar host-pathogen interaction (such as use of the same entry receptor as NL63, an α-hCoV) or similarities in the mode of action of host suppressive viral proteins. Interestingly, some longitudinal samples from Fig. 3a, showed increased hCoV response post SARS-CoV-2

exposure (Supplementary Fig. 7). However, to further explore cross-reactivity and correlation between CoV-induced immune responses, additional longitudinal samples from donors after SARS-CoV-2 infection are needed to generate meaningful conclusions.

## Discussion

We demonstrated that our MultiCoV-Ab, a multiplex immunoassay, is highly suitable to classify seroconversion in SARS-CoV-2-infected individuals. With a combined cut-off using SARS-CoV-2 trimeric full-length spike protein and RBD, we were able to eliminate false positive responses and achieved a sensitivity of 90% with a specificity of 100% for 310 samples from SARS-CoV-2-infected and for 866 samples from uninfected individuals. We found that detection of IgG more accurately reflected infection compared to IgA, although both were highly specific. However, by simultaneously monitoring IgA, we additionally were able to detect an early immune response in some patients. Interestingly, Yu, et al.[26] found that enhanced IgA responses might confer damaging effects in severe COVID-19. This is consistent with the observed significant increase in N protein directed IgA in hospitalized COVID19-cases, and confirms that careful monitoring of serum IgA warrants further attention.

The MultiCoV-Ab approach allows the easy addition of SARS-CoV-2-specific antigens, here 6 in total, which provides an additional level of confidence in patient classification. Thus, for example, we noticed that the spike S1 domain showed fewer false-positive responses compared to the S2 domain. Interestingly, Ng et al.[9] reported reactivity towards SARS-CoV-2 S2 from sera of patients with recent seasonal hCoV infection. These sera prevented infection with SARS-CoV-2 pseudotypes in a neutralization assay. Additionally, we found that spike non-responders also did not show a response to nucleocapsid but not vice versa, where nucleocapsid has been described as the strongest inducer of antibody responses[35,36]. Interestingly, nucleocapsid showed significant unspecific Ig binding in our assay. It has been previously reported, that the SARS-CoV-2 N protein is highly positively charged which may facilitate binding of viral nucleic acid but also result in unspecific binding of negatively charged molecules[37]. In addition, N protein oligomerization which is required to form the capsid[38] could further contribute to non-specific protein-protein interactions Therefore, our results highlight that the performance of an antigen is highly specific to the assay setup and cannot be easily generalized.

Another study measuring comparable numbers of serum/plasma samples using multiplex Luminex technology reports similar sensitivities and specificities for SARS-CoV-2 classification[20]. In our comparison to commercially available IVD tests, the MultiCoV-Ab classified fewer samples from SARS-CoV-2 infected donors as negative. However, for 10% of all infected samples, we could not detect a SARS-CoV-2 specific immune response, both in our measurement with MultiCoV-Ab and the commercial IVD kits. Intriguingly, others have already reported that up to 10% of SARS-CoV-2 patients do not develop detectable Ig levels[12,20,39]. Whether those non-responders are able to limit viral replication by innate immune mechanisms[40], forms of pre-existing immunity[41], or cellular immunity[42–44] is dominant in mediating viral clearance remains however to be determined.

One of the strengths of our study, compared to earlier studies[17,20,27], is the relatively large number of control and SARS-CoV-2 infected sera. However, a limitation is the potential bias introduced by an uneven age distribution across the study population. The uninfected control cohort was heavily skewed towards the age group of >60, whereas non-hospitalized COVID-

19 cases were over-represented in the age groups below 60. Despite this, MultiCoV-Ab specificity will still be accurate as all uninfected samples were identified correctly and all age groups were well represented with >100 samples per group.

Expanding our MultiCoV-Ab to the endemic hCoVs NL63, 229E, OC43, and HKU1 revealed a clear IgG immune response for all tested samples. Furthermore, we did not observe a difference for the samples from PCR-confirmed hCoV-infected individuals, compared to all others, suggesting that there is a significant degree of pre-exposure in the general population for all endemic hCoVs. Due to the general lack of availability of samples from hCoV-naive individuals, it was difficult to analyze hCoV-mediated cross-reactivity, set a cut-off and subsequently calculate specificities and sensitives for the hCoV S1, N-NTD, N antigens used here. Nevertheless, our multiplexed readout indicates a correlation between the SARS-CoV-2 immune response and high hCoV responses. Currently, we are identifying population groups which were highly exposed and showed different susceptibility to SARS-CoV-2 infection, e.g., the "Ischgl-study group" (unpublished data)[45], in order to elucidate potential cross-protection derived from immune responses towards endemic hCoVs in more detail. Alternatively, studies analyzing hCoV signatures in samples from individuals before and after SARS-CoV-2 infection using the MultiCoV-Ab would help to gain insight into a potential cross-protection.

A multiplex setup such as in MultiCoV-Ab is especially suited to vaccination studies, since the flexibility and broad antigen coverage allows to efficiently map vaccine immune responses to an immunoglobulin isotype and subtype level for the target pathogen and related species[17]. Interestingly, previous SARS-CoV-1 vaccine studies clearly indicated that a detailed characterization of vaccine-induced antibody responses is mandatory for efficient coronavirus vaccine development[46,47]. For instance, Yasui et al.[46] reported that although vector vaccines encoding SARS-CoV-1 S or N protein lead to comparable levels of anti-S and anti-N IgG in the respective study groups, N protein-immunized mice showed vaccine-induced pathology characterized by more severe lung damage, increased pulmonary neutrophil and eosinophil infiltration, and a significant upregulation of pro-inflammatory cytokine secretion upon challenge[43].

In summary, we have established and clinically validated the MultiCoV-Ab, a robust, high-content-enabled, and antigen-saving multiplex assay. This assay is suitable for comprehensive characterization of SARS-CoV-2 infection on the humoral immune response and for epidemiological screenings to accurately measure SARS-CoV-2 seroprevalence in large cohort studies. It could also provide the unique opportunity to assess and correlate immunity for both endemic and pathogenic coronaviruses. Finally, a broad and flexible antigen range through the multiplex nature of the MultiCoV-Ab can deliver urgently needed data to help guide decisions for SARS-CoV-2 vaccination strategies.

## Methods

**Generation of expression constructs for viral antigen production**. The sequence optimized cDNAs encoding the full-length nucleocapsid proteins of SARS-CoV-2, hCoV-OC43, hCoV-NL63, hCoV-229E, and hCoV-HKU1 (GenBank accession numbers "QHD43423.2"; "YP_009555245.1"; "YP_003771.1"; "NP_073556.1"; "YP_173242.1") were produced with an N-terminal hexahistidine (His$_6$)-tag by DNA synthesis (ThermoFisher Scientific). The cDNAs were cloned by standard techniques into NdeI/HindIII sites of the bacterial expression vector pRSET2b (ThermoFisher Scientific). The N-terminal domains (NTDs) of all nucleocapsid proteins were designed based upon previously published structural data[48]. Through this we were able to monitor the immune response against a rigid folded domain and exclude potential unspecific interactions with the largely unstructured region located between N-NTD and N-CTD of the nucleocapsid proteins. Furthermore, by depleting the N-CTD which is responsible for oligomerization of the nucleocapsid, we aimed to monitor antibody binding of a monomeric version of the

nucleocapsid. To generate NTDs of the respective nucleocapsid proteins (SARS-CoV-2 NTD aa 1-189; hCoV-OC43 NTD aa 1-204; hCoV-NL63 NTD aa 1 - 154; hCoV-229E NTD aa 1-156; hCoV-HKU1 NTD aa 1- 203), a stop codon located N-terminally to the Serine-Arginine (SR)-rich linker site[49] was introduced via PCR mutagenesis of the nucleocapsid encoding plasmids using the forward primer pRSET2b down-for and respective reverse primers: SARS-CoV2_NTD-rev, OC43_NTD-rev, NL63_NTD-rev, 229E_NTD-rev, and HKU1_NTD-rev.

Primer sequences are shown in Supplementary Table 4.

The pCAGGS plasmids encoding the stabilized trimeric Spike protein and the receptor binding domain (RBD) of SARS-CoV-2 were kindly provided by F. Krammer[27].

The cDNA encoding the S1 domain (aa 1–681) of the SARS-CoV-2 spike protein was obtained by PCR amplification using the forward primer S1_CoV2-for and reverse primer S1_CoV2-rev and the full length SARS-CoV-2 spike cDNA as template and cloned into the XbaI/NotI-digested backbone of the pCAGGS vector, thereby adding a C-terminal His$_6$-Tag.

The cDNAs encoding the S1 domains of hCoV-OC43 (aa 1–760), hCoV-NL63 (aa 1–744), hCoV-229E (aa 1–561) and hCoV-HKU1 (aa 1–755) (GenBank accession numbers "AVR40344.1"; "APF29071.1"; "APT69883.1"; "AGW27881.1") were produced by DNA synthesis (ThermoFisher Scientific), digested using XbaI/NotI and ligated into the pCAGGS vector. All expression constructs were verified by sequence analysis. An overview of all expressed constructs can be found in Supplementary Table 5.

**Protein expression and purification.** For the expression of the viral nucleocapsid proteins (full-length nucleocapsid and N-NTDs), the respective expression constructs were used to transform *E.coli* BL21 (DE3) cells. Protein expression was induced in 1 L TB medium at an optical density (OD$_{600}$) of 2.5–3 by addition of 0.2 mM isopropyl-β-D-thiogalactopyranoside (IPTG) for 16 h at 20 °C. Cells were harvested by centrifugation (10 min at 6,000 × g) and the pellets were then suspended in binding buffer (1x PBS, ad 0.5 M NaCl, 50 mM imidazole, 2 mM phenylmethylsulfonyl fluoride, 2 mM MgCl$_2$, 150 µg/mL lysozyme (Merck) and 625 µg/mL DNaseI (Applichem)). Cell suspensions were sonified for 15 min (Bandelin Sonopuls HD70 - power MS72/D, cycle 50%) on ice, incubated for 1 h at 4 °C in a rotary shaker followed by a second sonification step for 15 min. After centrifugation (30 min at 20,000 × g), urea was added to a final concentration of 6 M to the soluble protein extract. The extract was filtered through a 0.45 µm filter and loaded on a pre-equilibrated 1-mL HisTrap$^{FF}$ column (GE Healthcare). The bound His-tagged nucleocapsid proteins were eluted by a linear gradient (30 mL) ranging from 50 to 500 mM imidazole in elution buffer (1x PBS, pH 7.4, 0.5 M NaCl, 6 M urea). Elution fractions (0.5 mL) containing the His-tagged nucleocapsid proteins were pooled and dialyzed (D-Tube Dialyzer Mega, Novagen) against PBS.

The viral S1-domains, SARS-CoV-2 RBD, and the stabilized trimeric SARS-CoV-2 spike protein were expressed in Expi293 cells following the protocol as described in Stadlbauer et al.[19]. In brief, Expi293F-cells were cultivated (37 °C, 125 rpm, 8% (v/v) CO$_2$) to a density of 5.5 × 10$^6$ cells/mL. The cells were diluted with Expi293F expression medium to a density of 3.0 × 10$^6$ cells/mL, followed by transfection of the corresponding expression plasmids (1 µg per mL cell culture) with Expifectamine dissolved in Opti-MEM medium, according to the manufacturer's instructions. After 20 h post-transfection, transfection enhancers were added as documented in the Expi293F-cells manufacturer's instructions. The cell suspensions were cultivated for 2–5 days (37 °C, 125 rpm, 8% (v/v) CO$_2$) and centrifuged (4 °C, 23,900 × g, 20 min) to clarify the supernatant. The supernatants were filtered using a 0.22 µm membrane filter (Millipore, Darmstadt, Germany) and supplemented with His-A buffer stock solution (final concentration in the medium: 20 mM Na$_2$HPO$_4$, 300 mM NaCl, 20 mM imidazole, pH 7.4), before the solution was applied to a HisTrap FF crude column on a Äkta pure system (GE Healthcare, Freiburg, Germany). The columns were extensively washed with His-buffer-A (20 mM Na$_2$HPO$_4$, 300 mM NaCl, 20 mM imidazole, pH 7.4) before bound proteins were eluted with a imidazole gradient ranging from 50 mM – 400 mM. Eluted proteins were dialyzed against PBS and concentrated to 1 mg/mL.

All purified proteins were analyzed via standard SDS-PAGE followed by staining with InstantBlue Coomassie stain (Expedeon) and immunoblotting using an anti-His antibody (Penta-His Antibody, #34660, Qiagen, used at 1:1,000 dilution) in combination with a donkey-anti-mouse antibody labeled with AlexaFluor647 (#A31571, Invitrogen, used at 1:1,000 dilution) on a Typhoon Trio (GE-Healthcare, Freiburg, Germany; excitation 633 nm, emission filter settings 670 nm BP 30) to confirm protein integrity. To further confirm correct expression, integrity, and purity, proteins were analyzed by mass spectrometry. To control the production reproducibility of the antigens, potential aggregation and melting temperatures of the proteins were investigated by nano differential scanning fluorimetry (nanoDSF) using a Prometheus (Nanotemper, Munich, Germany).

**Commercial antigens.** Two commercial antigens were used to complement the in-house-produced antigen panel.

The S2 ectodomain of the SARS-CoV-2 spike protein (aa 686–1213) was purchased from Sino Biological, Eschborn, Germany (cat # 40590, lot # LC14MC3007). A full-length nucleocapsid protein of SARS-CoV-2 was purchased from Aalto Bioreagents, Dublin, Ireland (cat # 6404-b, lot # 4629).

**Bead-based serological multiplex assay.** All antigens were covalently immobilized on spectrally distinct populations of carboxylated paramagnetic beads (MagPlex Microspheres, Luminex Corporation, Austin, TX) using 1-ethyl-3-(3-dimethylaminopropyl)carbodiimide (EDC)/sulfo-N-hydroxysuccinimide (sNHS) chemistry. For immobilization, a magnetic particle processor (KingFisher 96, Thermo Scientific, Schwerte, Germany) was used.

Bead stocks were vortexed thoroughly and sonicated for 15 s. Subsequently, 83 µL of 0.065% (v/v) Triton X-100 and 1 mL of bead stock containing 12.5 × 10$^7$ beads of one single bead population were pipetted into each well. The beads were then washed twice with 500 µL of activation buffer (100 mM Na$_2$HPO$_4$, pH 6.2, 0.005% (v/v) Triton X-100) and beads were activated for 20 min in 300 µL of activation mix containing 5 mg/mL EDC and 5 mg/mL sNHS in activation buffer. Following activation, the beads were washed twice with 500 µL of coupling buffer (500 mM MES, pH 5.0, 0.005% (v/v) Triton X-100) and the antigens were added to the activated beads and incubated for 2 h at 21 °C to immobilize the antigens on the surface.

Antigen-coupled beads were washed twice with 800 µL of wash buffer (1x PBS, 0.005% (v/v) Triton X-100) and were finally resuspended in 1000 µL of storage buffer (1x PBS, 1% (w/v) BSA, 0.05% (v/v) ProClin). The beads were stored at 4°C until further use.

To detect human IgG and IgA responses against SARS-CoV-2 and the endemic human coronaviruses (hCoV-NL63, hCoV-229E, hCoV-OC43 and hCoV-HKU1), the purified trimeric spike protein (S), S1-domain, S2-domain (Sino Biological GmbH, Europe), RBD, nucleocapsid (N) and the N-terminal domain of nucleocapsid (N-NTD) of SARS-CoV-2 as well as the S1-domain, N, and N-NTD of the endemic hCoVs were immobilized on different bead populations as described above. The individual bead populations were combined into a bead mix. A bead-based multiplex assay was performed. Briefly, samples were incubated at a 1:400 dilution for 2 hours at 21 °C. Unbound antibodies were removed and the beads were washed three times with 100 µL of wash buffer (1x PBS, 0.05% (v/v) Tween20) per well using a microplate washer (Biotek 405TS, Biotek Instruments GmbH). Bound antibodies were detected with R-phycoerythrin labeled goat-anti-human IgG (Dianova, Cat# 109-116-098, Lot#148837, used at 3 µg/mL) or IgA (Dianova, Cat# 109-115-011, Lot#143454, used at 5 µg/mL) antibodies (incubation for 45 min at 21 °C). For each sample, a single measurement was performed. Readout was done using a Luminex FLEXMAP 3D instrument and the Luminex xPONENT Software 4.3 (settings: sample size: 80 µL, 50 events, Gate: 7,500–15,000, Reporter Gain: Standard PMT).

**Quality control and technical assay validation steps.** In order to test the repeatability of the MultiCoV-Ab three quality control samples (QCs) were processed in duplicate on each test plate (n = 17) during the sample screening and inter-assay variance was assessed for each antigen in the multiplex. For intra-assay variance, 24 replicates for each of the three QC samples were analyzed on one plate. Results from this are presented in Supplementary Table 1 and Supplementary Fig. 2. A limit of detection (LOD) for each antigen was determined by processing a blank in 24 replicates and the LOD was set as mean MFI + 3 standard deviations. Sample parallelism and comparability of paired serum and plasma samples were assessed over eight dilution steps ranging from 1:100 to 1:12,800 (Supplementary Fig. 2). A set of samples derived from 205 SARS-CoV-2-infected and 72 uninfected individuals was tested repeatedly with two different kit batches. The samples classification in both runs matched 100%. Furthermore, as part of our negative sample panel, we have analyzed samples with potentially interfering characteristics (i.e., samples from patients with PCR-confirmed hCoV infection, presences of HAMA (human anti-mouse antibodies) and rheumatoid factor (RF), with high procalcitonin values (> 3 ng/mL), as well as from pregnant women and patients with neuroinflammatory diseases) (Supplementary Table 2).

**Samples.** A total of 1176 sera and plasma samples were used for the MultiCoV-Ab assay development. Ethical approval was granted from the Ethics Committee of Hannover Medical School (#9122_BO_K2020). Only de-identified samples were used for the MultiCoV-Ab assay development. All samples were pre-existing. Cohort age was 5-88 years; age was not known for 161 samples.

310 samples were from COVID-19 patients or convalescents. Samples were classified as SARS-CoV-2 infected, if a positive SARS-CoV-2 RT-PCR was reported and/or if hospitalization/quarantine for COVID-19 was indicated as part of the samples metadata. dT defined as time between PCR test or symptom onset and blood draw was 0–73 days (median= 38 d; n = 258). dT was not provided for 52 samples. SARS-CoV-2 infected samples used in this study were collected after ethical review (9001_BO_K, Hannover Medical School; 179/2020/BO2, University Hospital Tübingen; 85/20, Ärztekammer des Saarlandes).

866 control samples were from non-SARS-CoV-2 infected individuals and were classified as non-infected as they were obtained prior to the emergence of SARS-CoV-2 in December 2019 or because they were taken from individuals who had not reported cold symptoms since the beginning of 2020.

The majority of non-SARS-COV-2 infected samples were randomly selected and consisted of pre-pandemic blood donors, commercially available (Central BioHub GmbH, Berlin, Germany and BBI Solutions, Crumlin, UK) or bio-banked specimens. 365 samples were from the Memory and Morbidity in Augsburg Elderly (MEMO) study (a sub-cohort of the MONICA S2 cohort (WHO 1988)) and were included based on available serological titers for HSV-1, HSV-2, HHV-6, and

EBV[50]. 88 samples were obtained from transplanted patients with chronic respiratory conditions.

Collection of non-SARS-CoV-2 infected control samples had been approved by several ethic committees: 3232-2016 (Ethics Committee of Hannover Medical School); 62/20 (Ethics Committees of the Medical Faculty of the Saarland University at the Saarland Ärztekammer); WUM 17.02.1997 (Joint ethics committee of the University of Münster and the Westphalian Chamber of Physicians).

All necessary patient/participant consent has been obtained and the appropriate institutional forms have been archived. Additional sample details can be found in Supplementary Table 2. Serum and plasma samples were handled in Class II-laminar flow benches in L2 laboratories[51]. Samples were not heat-inactivated. All incubation steps took place in fully sealed assay plates.

**Data analysis**. Data analysis and visualization were performed with R Studio (Version 1.2.5001, using R version 3.6.1) using the Median Fluorescent Intensity (MFI). Statistical analysis was performed using R package "stats" from the base repository. Mann-Whitney U test was used to determine the difference between signal distributions from different sample groups. Spearman's $\rho$ coefficient was calculated in order to correlate antigens by response from the entire sample set, followed by hierarchical clustering to group antigens. Fishers' exact test was used to calculate the significance of overlap between sample groups. 95% Confidence intervals for sensitivity and specificity values calculated in this study were calculated after Clopper-Pearson[52] and associated positive and negative predictive values (PPV/NPV) were calculated based on a seropositivity of 3%. Sequence alignments and sequence identity scores were calculated with version 1.2.4. of Clustal Omega[53].

**Reporting summary**. Further information on research design is available in the Nature Research Reporting Summary linked to this article.

## Data availability
Data relating to the findings of this study are available from the corresponding author upon request. Source data have been deposited on GitHub alongside the analysis code: https://github.com/BeckerMatthias/MULTICOV-AB_Publication/. Source data are provided with this paper.

## Code availability
Analysis code and required input files have been deposited on GitHub: https://github.com/BeckerMatthias/MULTICOV-AB_Publication/

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

## Acknowledgements

This work was supported by the Initiative and Networking Fund of the Helmholtz Association of German Research Centres (Grant number SO-96). This work has further received funding from the European Union's Horizon 2020 research and innovation programme under Grant agreement no. 101003480–CORESMA. We thank Florian Krammer for providing us with expression plasmids for the Spike Trimer and RBD. We thank Shannon Layland for critically proofreading of the manuscript.

## Author contributions

M.B. designed and performed experiments and data analysis; M.S. designed experiments; D.J., J.H., S.F., F.R., planned and performed experiments; A.Z. performed mass spectrometry analysis; J.H. performed nanoDSF analyses; H.D., B.T., P.D.K., F.W., U.R. designed, cloned, expressed and purified the antigens; S.H. and A.P. performed sample analysis; T.B., A.B., S.L., S.S., M.C., T.I., J.G., A.K., K.B., H-G.R., A.N., M.M., J.S.H., J.S.W., M.T., T.O.J. arranged sample and data collection; K.S-L., M.T., T.O.J, T.K., G.K. supported the study planning; S.G. reviewed the analysis code; N.S-M. planned the study, assay development, and validation and designed experiments; M.B., M.S., A.D., U.R., and N.S-M. wrote the manuscript. All authors reviewed the manuscript.

## Competing interests

The authors declare the following competing interests: T.O.J. is a scientific advisor for Luminex. N.S-M. was a speaker at Luminex user meetings in the past. The Natural and Medical Sciences Institute at the University of Tübingen is involved in applied research projects as a fee for services with Luminex. The remaining authors declare no competing interests.

## Additional information

[1]NMI Natural and Medical Sciences Institute at the University of Tübingen, Reutlingen, Germany. [2]Department of Epidemiology, Helmholtz Centre for Infection Research, Braunschweig, Germany. [3]TWINCORE GmbH, Centre for Experimental and Clinical Infection Research, a joint venture of the Hannover Medical School and the Helmholtz Centre for Infection Research, Hannover, Germany. [4]Helmholtz-Institute for RNA-based Infection Research (HIRI), Würzburg, Germany. [5]Pharmaceutical Biotechnology, University of Tübingen, Tübingen, Germany. [6]Institute for Clinical Chemistry and Pathobiochemistry, Department for Diagnostic Laboratory Medicine, University Hospital Tübingen, Tübingen, Germany. [7]Institute for Diabetes Research and Metabolic Diseases of the Helmholtz Center Munich at the University of Tübingen, Tübingen, Germany. [8]German Center for Diabetes Research (DZD), München-Neuherberg, Germany. [9]Institute for Clinical and Experimental Transfusion Medicine, University Hospital Tübingen, Tübingen, Germany. [10]Niedersächsisches Landesgesundheitsamt, Department of Virology/Serology, Hannover, Germany. [11]Institute of Virology, Saarland University Medical Center, Homburg/Saar, Germany. [12]Department of Gastroenterology, Hepatology, Endocrinology, Hannover Medical School, Hannover, Germany; Centre for Individualized Infection Medicine (CiiM), Hannover, Germany. [13]Hannover Unified Biobank (HUB), Hannover Medical School, Hannover, Germany. [14]Department of Respiratory Medicine, Hannover Medical School, Hannover, Germany. [15]Biomedical Research in End-stage and obstructive Lung Disease Hannover (BREATH), Member of the German Center for Lung Research (DZL), Hannover, Germany. [16]Institute of Epidemiology and Social Medicine, University of Münster, Münster, Germany. [17]Institute for Cell Biology, Department of Immunology, University of Tübingen, Tübingen, Germany. [18]German Cancer Consortium (DKTK) and German Cancer Research Center (DKFZ), partner site Tübingen, Tübingen, Germany. [19]Cluster of Excellence iFIT (EXC2180) "Image-Guided and Functionally Instructed Tumor Therapies", University of Tübingen, Tübingen, Germany. [20]Department of Women's Health, Research Institute for Women's Health, Eberhard-Karls-University, Tübingen, Germany. [21]Department of Medicine/Cardiology, Cardiovascular Research Laboratories, David Geffen School of Medicine at UCLA, Los Angeles, CA, USA. [22]Clinical Collaboration Unit Translational Immunology, German Cancer Consortium (DKTK), Department of Internal Medicine, University Hospital Tübingen, Tübingen, Germany. [23]Dr. Margarete Fischer-Bosch Institute of Clinical Pharmacology (IKP) and Robert Bosch Center for Tumor Diseases (RBCT), both Stuttgart, Germany. [24]These authors contributed equally: Matthias Becker, Monika Strengert, Ulrich Rothbauer. ✉email: Nicole.Schneiderhan@nmi.de

