## [Peer Review File · Nature Communications]

Reviewers' Comments:

Reviewer #1:

Remarks to the Author:

In this manuscript the authors describe a multiplex methodology for the detection of SARS CoV-2 proteins (Spike, S1, S2, NTD and N proteins) and circulating HCoV-2 (S1 and N proteins from NL63, OC43, 229E and HKU1). These multiplex assays have frequently appeared in the literature in the last 6 months or so, and demonstrate utility including immune surveillance and guiding vaccine strategy. Becker et al initially align their assay with 3 clinical assays used in diagnosis then move forward in testing a rich dataset of individuals for IgA and IgG, including a nicely conducted longitudinal analysis. In the last part of the analyses the group tested the hypothesis of cross reactivity with circulating HCoV-2 antigens.

Overall the data in the manuscript is extremely interesting and thought provoking but the authors miss an opportunity to elaborate on some of the nuisances within the data. For example Figure 1b; it is interesting that commercially available clinical diagnostic assays target, either or both, N or S antigens, which could be accounted for in correlative analyses or commented upon from their initial dataset. Are the inconsistencies seen between the assays due to antigen selection? The discordant rates between assays seem quite distinctive in this dataset, which seems worryingly typical for these datasets for CoV-2 diagnoses.

However, in general it seems the manuscript would be clearer if the authors addressed the following:

- Common in these assays to have a spread of reactivities to CoV-2 spike, which are potentially associated with severity or even timing post infection, therefore:
 - o What PCR test defined the CoV-2 positive and negative samples
 - o Time post diagnosis of the CoV-2 positive and negatives – waning antigen or antibody issues?
- Does having more than 1 antigen increase sensitivity and specificity compared to a single antigen
- Details of how the actual cut-offs were calculated in the main text and how this was applied to the second dataset
- Speculations of the consistent overlap between CoV-2 positive and negatives?
- High cross reactivity for N proteins in uninfected samples?
- Expand further on relationships between high CoV-2 Spike and those from circulating HCoV?
- Would figure #1 maybe better represented by a scatterplot?
- Maybe figure 3b would be better represented in the main text with gender and age?
- Speculation about the relative IgG binding for the domains and implications for further testing of CoV-2 spike immunized volunteers?
- Explanation or postulation on why antibody levels for circulating HCoV spike protein differ from N protein and is this difference significant clinically?
- definition of “validated”, which is a much-used term in academia and has a different connotation in industry settings, maybe replace by “optimized” to keep nomenclature consistent? Unless this is really validated? If so, then please expand?
- o Perhaps consider a standard curve and trending positive and negative controls over each run for these analyses
- Slight confusion about breaking figure 4 into clades and summing the data, consider explaining or graphically representing in a clearer manner
- Would give a better insight into the utility of these binding assays if there were a measure of ACE2 target recognition and/or functional neutralization analyses

Reviewer #2:

Remarks to the Author:

The authors describe the development and initial testing of a SARS-CoV-2 antibody test. This antibody test can detect IgG and IgA antibodies raised against the SARS-CoV-2 spike or RBD proteins, but is

also able to measure responses against other proteins like the nucleoprotein or other subunits (S1, S2) of the spike. In addition, in this immunoassay the detection of antibodies reactive to the spike and nucleoprotein of human seasonal coronavirus is being done in a multiplex manner. Leveraging a sample set derived from subjects with diagnosed SARS-CoV-2 infection and a set of negative control samples the sensitivity and specificity of the assay is established. Using multiple antigens for determination and classification of positivity the assay achieves 100% specificity. The authors report high levels of antibodies binding to proteins of human seasonal coronavirus, but cannot, with the available set of samples, detect clear cross-reactive responses in samples from subjects that were infected with SARS-CoV-2.

The submitted manuscript describes in detail how this novel assay was being set-up and show a high level of reproducibility and stability in their assay. Moving forward this test can be used to measure the antibody response towards many antigens simultaneously and help answer important questions, however, below are a few comments that should be addressed.

Major

- The abstract is missing. I think both an abstract and an introduction are needed.
- Some papers report cross-reactive responses that target the S2 domain of coronavirus spike proteins (both SARS-CoV-2 and hCoVs). Antibodies that target this region of hCoV spikes are not being captured by the MultiCoV-Ab immunoassay since only the S1 region is being probed. Is there a plan to include the S2 domain as an antigen in the assay?
- Please add the sensitivity/specificity metrics for the commercial IVD tests used if available. Is the reported sensitivity of these assays in line with the results found in this study?
- Please rephrase line 95 and clarify that all 24 false negatives in the MultiCoV-Ab were also false negatives in the commercial assays. What is known about the samples from infected subjects used for Figure 1? Is there an explanation for the 24 samples that were incorrectly classified as negative (e.g. time between infection and blood draw for serology)? Is there a positive PCR result for all 205 subjects? The issue of timing and lag between infection to generation of measurable antibody levels could also be mentioned in the discussion (line 196-200). Please also mention the sensitivity and specificity of the assay (including statistics) for this first set of tested samples.
- In Figure 1b all samples from SARS-CoV-2 infected people show up as seropositive as measured with MultiCoV-Ab, however in the text it is mentioned that 24 samples were IgG negative. I think it would also be good to include numbers (x positive / y samples from infected subjects) in the panel in Figure 1b to increase clarity. Further, it seems like that the sample numbers (infected/non-infected) differ when testing for IgA, please clarify
- Please include the confidence interval for the sensitivities and specificities reported. Additionally, the positive and negative predictive values of the assay should be reported.
- How were the cut-off values for spike and RBD calculated and defined? Please add information about this process (line 115).
- SARS-CoV-2 belongs to beta coronavirus genera, however in Figure 4a the correlation of the IgG response clusters more closely to the alpha coronaviruses. This clustering is rather unexpected and may be caused by differences in pre-existing antibody levels to human coronaviruses. Also, it would be of interest to the reader to see a sequence alignment to be able to compare how similar/different the proteins used in this study are (e.g. NL63 S1 vs OC43 S1 vs SARS-CoV-2 S1 etc.)
- Line 168-178: Human coronaviruses are viruses that are circulating seasonally and every adult has been exposed to those viruses multiple times. Based on this pre-exposure history and recent

infections subject will have high or low antibody levels toward hCoV S1 and N. Hence, the grouping in high and low responders is likely based on infection history and might or might not be influenced by SARS-CoV-2 infection. A statement should be included to discuss pre-exposure with human seasonal coronaviruses.

Minor

- Line 73: Please rephrase; the highest sensitivity would be 100. How does the reported sensitivity of 90% compare to other Luminex based immunoassays?
- Please indicate the cut-off (in MFI) for spike and RBD in Figure 1a. Similarly, a line for the LOD for each antigen would be helpful.
- Line 103: Please add "when testing for IgA antibodies in serum/plasma"
- The numbers in the table in Figure 2a seem slightly misaligned.
- Is the ROC analysis that is mentioned in line 113 shown somewhere?
- In Line 119 the authors mention that IgA responses were measured to capture early immune responses after infection, however, typically IgM antibodies are detectable first followed by IgA and IgG. Accordingly, in Figure 3a a similar profile for IgG and IgA can be seen. Please clarify and/or add information.
- It would be nice to have the same y-axis scale for IgG and IgA to be able to easily see and compare that the IgA response is lower than IgG in those patients. In figure 3b and c the left and right y-axis scales should be aligned to allow for better comparison. The respective cut-off values should be indicated. The same goes for figure 4b and c where different scales are being used.
- How does the commercially available full-length nucleoprotein compare to the in-house produced NP? Is the expression system similar or is there a possible difference in glycosylation pattern that might influence antibody binding?
- Please add which safety precautions were taken when dealing with serum/plasma (e.g. heat inactivation of serum samples from infected subjects) in the methods section.
- Out of curiosity, how many of the alpha-hCoV high responders are also beta-hCoV high responders?
- It would be interesting to see longitudinal data for all antigens (also hCoV) tested for the five subjects shown in Figure 3a. If possible, a supplemental figure could be added. In general, looking at longitudinal samples will be important to help identify possible cross-reactive responses among human seasonal coronaviruses and the pandemic virus.

Reviewer #3:

Remarks to the Author:

The manuscript by Becker et al. describes the development of a multiplex CoV serology methodology including antigens from SARS-CoV-2 as well as endemic human CoVs. I agree with the authors that there is an urgent need for a well performing and easily applicable serological assay to determine sero-prevalence rates during the current pandemic in large sample sets. However, I have some major concerns regarding the study population used to validate the assay as well as conclusions drawn from the results.

Point-by-point comments to the manuscript are given below.

Introduction:

- The introduction in general is very brief and does not provide sufficient information to justify certain aspects of the current analyses (eg the measurement of anti-IgG and anti-IgA responses or the inclusion of other hCoVs). Furthermore, there is very little description of currently available and applied serologies in the literature. Also, the last paragraph is rather a summary of the results and already a conclusion of the performed work instead of pointing out the aim of the current study.

Methods:

- Please explain and provide references for why you chose to express the N-terminal domain of protein N

- A table with information on the applied primers and constructs would provide a great overview for the reader
- The selection of controls may introduce some bias into the results and should be described more detailed as well as discussed as a limitation. Study information given in Figure 2a and extended table 2 should include percentages to make differences in age and sex distribution easily accessible to the reader. As it seems from the numbers there is indeed a substantial age difference between cases and controls. Please discuss how this could have affected obtained results for sensitivity and specificity, especially since figure 3c shows an association of antibody response with age. Also, different sources of control samples and potentially resulting impact should be discussed.

Results:

- Please provide subheadings to facilitate reading and understanding for the reader.
- You describe a pilot with a smaller sample set in the results, were these samples later included in the larger sample set or handled separately?
- Please provide a reference on why IgA might indicate an early immune response.
- Is there any explanation as to why antibody responses to SARS-CoV-2 proteins correlate more strongly with alpha-CoVs than beta? Did you check the amino acid sequence homology?

Discussion:

- You discuss that 10% of the SARS-CoV-2 cases were not identified as sero-positives with your assay. How long after symptom onset were these samples drawn? Could this be an explanation for the lack of sensitivity?
- You describe correlations between responses to proteins of SARS-CoV-2 and those of other human CoVs. However, there is no data provided that shows that proteins of other human CoVs actually detect a past infection with the respective virus in a specific and sensitive way. Thus, I believe the conclusion that this assay provides the opportunity to measure an antibody response to these other hCoVs is over-interpreted and should be toned down.
- Also, the conclusion that the newly developed assay will be helpful in determining outcome of vaccination is over-interpreted in my opinion. There is no information given on how measured antibody responses relate to immunity against a re-infection or how this correlates with neutralization titers.

Point-to-point reply to reviewer's comments

Reviewer #1:

Comment #1:

Overall the data in the manuscript is extremely interesting and thought provoking but the authors miss an opportunity to elaborate on some of the nuisances within the data. For example Figure 1b; it is interesting that commercially available clinical diagnostic assays target, either or both, N or S antigens, which could be accounted for in correlative analyses or commented upon from their initial dataset. Are the inconsistencies seen between the assays due to antigen selection? The discordant rates between assays seem quite distinctive in this dataset, which seems worryingly typical for these datasets for CoV-2 diagnoses.

Author response:

We thank the reviewer for their overview of our work. We are convinced that the discrepancies between the commercial IVD tests are due to a combination of factors, including antigen selection (Euroimmun and Siemens use Spike-derived antigens, Roche uses a nucleocapsid-derived antigen) and assay type (Siemens and Roche are bridging assays and perform better than the Euroimmun test which is an ELISA).

Comment #2:

• Common in these assays to have a spread of reactivities to CoV-2 spike, which are potentially associated with severity or even timing post infection, therefore:

o What PCR test defined the CoV-2 positive and negative samples

Author response:

As stated in the manuscript (lines 631-664), our samples were obtained from several sources. Although we requested information about which PCR test was used, it was not always possible to obtain detailed information on the PCR performed for our complete sample cohort. We agree with the reviewer that this information would have been particularly interesting, as it would have allowed us to rule out that the antibody negative but PCR positive samples were due to a borderline low Ct value indicating a false positive PCR result.

o Time post diagnosis of the CoV-2 positive and negatives – waning antigen or antibody issues?

Author response:

We thank the reviewer for pointing out that we had failed to address this in the original manuscript. To address, we have now included an additional figure (Extended Data Figure 5), in which sampling time and its effects on seroconversion is described. We divided our samples into three groups (sampling of sera relative to symptom onset, PCR date or result of PCR) based on the available metadata. For IgG, we see that the antibody levels remain consistent, whereas a decrease with time post SARS-CoV-2 infection is observed for IgA. We expand upon and explain these findings in the results section (lines 216 to 218 and 221 to 227)

Comment #3:

• *Does having more than 1 antigen increase sensitivity and specificity compared to a single antigen*

Author response:

This was already included in the original manuscript. For clarity purposes and in response to other reviewers' comments, Figure 2 (which already contain this information) has been expanded and clarified. As stated in the manuscript, the combination of RBD and Spike Trimer results in a specificity of 100% while retaining an acceptable level of sensitivity (lines 180 to 182). Since we use a classification where 2 cut-offs have to be met, there is also a small loss of sensitivity compared to single antigens (Figure 2b).

Comment #4:

• *Details of how the actual cut-offs were calculated in the main text and how this was applied to the second dataset*

Author response:

We have expanded and rewritten for clarity our description of how cut-offs were calculated in the manuscript (lines 174 to 193). As now stated in the manuscript (line 171 to 174), cut-offs were calculated using the extended dataset (Fig 2). These cut-offs were then used for all SARS-CoV-2 classifications contained within this manuscript. With the expanded results section, we have added a figure for the performed ROC analysis (Extended Data Figure 4).

Comment #5:

- *Speculations of the consistent overlap between CoV-2 positive and negatives?*

Author response:

We thank you for this comment and the chance to comment further on this question. We have included further literature in this topic as part of our revised discussion (line 301 to 307), which now reads "..., for 10% of all infected samples, we could not detect a SARS-CoV-2 specific immune response, both in our measurement with MultiCoV-Ab and the commercial IVD kits. Intriguingly, others have already reported that up to 10 % of SARS-CoV-2 patients do not develop detectable Ig levels. Whether those non-responders are able to limit viral replication by innate immune mechanisms, forms of pre-existing immunity, or cellular immunity is dominant in mediating viral clearance remains however to be determined."

Comment #6:

- *High cross reactivity for N proteins in uninfected samples?*

Author response:

We speculate that the high cross-reactivity that was observed is tied to the N proteins function as a RNA binding protein and as a major component of the viral capsid. It further confirms that antigen performance is specific to the assay setup. We have addressed this in the revised manuscript in lines 207 to 209 and 291 to 297.

Comment #7:

- *Expand further on relationships between high CoV-2 Spike and those from circulating HCoV?*

Author response:

We did not observe any general correlation between reactivity to the SARS-CoV-2 Spike Trimer or the RBD with hCoV S1 proteins. We think our analysis (Figure 4, lines 230 to 269) is fairly conclusive for this topic. As stated within the manuscript, we clustered all antigens (Figure 4a) and then examined immune responses towards hCoV S1 proteins for both infected and uninfected samples (Figure 4b).

We then further investigated the hCoV S1 response for only uninfected samples, which were cross-reactive to SARS-CoV-2 Spike Trimer (Figure 4d). Our in-detail analysis of this can be found as lines 239 to 250. We have also included an alignment of the used S1 constructs to highlight sequence similarities (Extended Data Figure 7) as per another reviewer's request.

Comment #8:

- *Would figure #1 maybe better represented by a scatterplot?*

Author response:

We were slightly unsure if the reviewer was referring to Figure 1a or b with this comment. We believe that displaying Figure 1a as a scatterplot would be unproductive, as it would end up as a reduced version of Figure 2c, i.e. same plot with fewer samples. As Figure 1 shows the comparison amongst our assay and commercial IVD assays, and the performance of SARS-CoV-2 antigens with the extended sample set is already shown as Figure 2, we think it would be unnecessary. The data of Figure 1b are already displayed as scatterplots in Extended Data Figure 3.

Comment #9:

- *Maybe figure 3b would be better represented in the main text with gender and age?*

Author response:

We thank the reviewer for this comment. We have now expanded our results section on the dynamics of antibody response in COVID-19 patients, going into more detail about trends within our sample cohort (lines 221 to 227). Additionally, in response to comment #2, we have added an extended data figure 5 addressing effects for time post infection as well which further supplements this analysis. Overall, we do see effects with hospitalisation, age and time post infection, however our sample set was not designed to single out the leading cause amongst these trends.

Comment #10:

- *Speculation about the relative IgG binding for the domains and implications for further testing of CoV-2 spike immunized volunteers?*

Author response:

We did observe a greater number of reactive samples from uninfected donors for the S2 domain compared to the S1 (Figure 2e), which is consistent when comparing Spike Trimer, which includes S2 and RBD (Figure 2c). It appears that cross-reactivity is tied to the S2 domain, which in turn suggests that the S1 domain is the more specific one. We have addressed this in the results section in lines 196 to 203. However, we are hesitant to speculate which spike domain might be more suitable as a vaccine candidate based on this observation. Samples from actual vaccination studies would be required to properly answer this question.

Comment #11:

• *Explanation or postulation on why antibody levels for circulating HCoV spike protein differ from N protein and is this difference significant clinically?*

Author response:

We thank the reviewer for this comment. It should be noted that from a technical point of view, MFI signals may not be directly compared to one another for different antigens. Nevertheless, we observed and discussed these differences using the rank correlation analysis in Fig 4a (lines 232 to 236). We could speculate that differences between the N and S immune response could be caused by the degree of lytic cell death an individual experiences throughout the course of viral infection, in combination with the fact that the N protein is the most abundant protein in the viral particle. The resulting massively increased amounts of available N protein antigen could therefore create a skew towards N-protein directed antibodies in these individuals. We have now included a sentence on potential clinical significance of anti-S and anti-N response (lines 334 to 339). We do not want to speculate further on the clinical relevance of N-protein antibodies in infections with hCoV, due to the lack of relevant samples.

Based upon your comment, we also investigated whether there are any effects in the hCoV immune response for COVID-19 hospitalized vs non-hospitalized, but as expected from the results we already showed in Figure 4 b-c, we did not see any effects.

Comment #12:

• *Definition of “validated”, which is a much-used term in academia and has a different connotation in industry settings, maybe replace by “optimized” to keep nomenclature consistent? Unless this is really validated? If so, then please expand?*

Author response:

We thank the reviewer very much for this comment and agree that the word validation can be misleading. We therefore have altered our manuscript to clarify this and propose to use the wording “clinically validated” for our discussion instead, to showcase that we assembled a broad sample set to determine assay performance.

In terms of technical validation of the assay, we have assessed intra- and inter-assay variance and limit of detection, dilution linearity and comparability of serum and plasma samples (Extended Data Figure 2 and Extended Data Table 1). However, we have not yet tested for reagent stability, influence of freeze-thaw cycles on the sample response or for interference factors according to FDA and EMA guidelines.

We have expanded upon our validation strategy within the manuscript in the Methods section (line 613 to 629) and Results (line 138 to 140).

Comment #13:

o Perhaps consider a standard curve and trending positive and negative controls over each run for these analyses

Author response:

We would not like to include a standard curve, as due to the multiplex setup we would probably have to include multiple curves on every run, dramatically limiting the throughput capabilities of the assay. Instead we have now included control samples including cut-off values for every assay run. For the dataset presented in the manuscript, we have included the positive and negative control data for each run (Extended Data Table 1 and Extended Figure 2).

Comment #14:

• *Slight confusion about breaking figure 4 into clades and summing the data, consider explaining or graphically representing in a clearer manner*

Author response:

SARS-CoV-2 belongs to the clade of β -coronaviruses and as such is more closely related to other β -hCoVs in sequence. This is also supported by the newly added sequence alignments of our antigens in Extended Data Figure 7. The analysis in Figure 4a, implies cross reactivity among the antigens within one clade. We were therefore especially interested in the analysis of the different clades in follow up analyses, hence we split the samples into high and low responders within a clade for Figure 4f. We are convinced that the analyses progression is logical in its current form. Thus, we decided to leave Figure 4 unchanged. We have added this reasoning in the text to make the analysis easier to follow (lines 250 to 253) and expanded on the interpretation of Fig 4a (lines 233 to 236).

Comment #15:

• *Would give a better insight into the utility of these binding assays if there were a measure of ACE2 target recognition and/or functional neutralization analyses*

Author response:

We are already establishing an assay to measure the content of neutralising antibodies towards SARS-CoV-2, independently of in vitro neutralisation assays, for which first results have recently been released in a pre-print (<https://doi.org/10.1101/2020.09.22.308338>).

Reviewer #2:**Comment #1:**

- *The abstract is missing. I think both an abstract and an introduction are needed.*

Author response:

We thank the reviewer for this comment. As the original submission of the manuscript was as a letter, we had not included these sections as per the editorial guidelines. As this manuscript is now a full article, we have as such updated the manuscript and included both an abstract and an introduction within the revised manuscript.

Comment #2:

- *Some papers report cross-reactive responses that target the S2 domain of coronavirus spike proteins (both SARS-CoV-2 and hCoVs). Antibodies that target this region of hCoV spikes are not being captured by the MultiCoV-Ab immunoassay since only the S1 region is being probed. Is there a plan to include the S2 domain as an antigen in the assay?*

Author response:

We thank the reviewer for this comment. We fully agree that the S2 domain is a highly interesting target and may be especially relevant for cross-reactivity of coronaviruses. As such, we are currently working to expand our hCoV antigen panel to include S2 domains and trimeric full-length Spike proteins.

Comment #3:

- *Please add the sensitivity/specificity metrics for the commercial IVD tests used if available. Is the reported sensitivity of these assays in line with the results found in this study?*

Author response:

As per the reviewer's comment, we have now expanded Figure 1 to include a list of found sensitivity and specificity with confidence intervals, as well as manufacturer specifications. We have also expanded the discussion of assay performance, comparing measured results of samples in Figure 1 to the performance data from the assay specification sheets in lines 163 to 168.

Comment #4:

• *Please rephrase line 95 and clarify that all 24 false negatives in the MultiCoV-Ab were also false negatives in the commercial assays. What is known about the samples from infected subjects used for Figure 1? Is there an explanation for the 24 samples that were incorrectly classified as negative (e.g. time between infection and blood draw for serology)? Is there a positive PCR result for all 205 subjects? The issue of timing and lag between infection to generation of measurable antibody levels could also be mentioned in the discussion (line 196-200). Please also mention the sensitivity and specificity of the assay (including statistics) for this first set of tested samples.*

Author response:

We thank the reviewer for this comment and suggestions on improving the clarity of the manuscript. In response, we have rephrased line 95 (now line 151 to 153) to state “Of the 205 infected samples, both MultiCoV-Ab assay and commercial IVD tests for total Ig or IgG identified 24 (11.7%) as IgG antibody-negative”. For the initial set of samples, all 205 SARS-CoV-2 infected donors were convalescent, meaning that if an antibody response was developed, it should have been measurable. This is now stated in the manuscript (line 143). Unfortunately, the 205 samples are self-reported positives, which means we do not have access to data such as Ct values or PCR assay used. We have addressed the timing of sampling in an additional figure (Extended Data Figure 5) which feeds into the results for Figure 3, where we examined the effects of Age and Hospitalisation. In our analyses, there was no trend among the available metadata for the aforementioned 24 “false negative” samples. We also investigated those exact samples for response to hCoV antigens and did not see a general trend there either (Figure 4e-f). As mentioned in response to comment #3, we expanded Figure 1 to include the statistics for the compared assays.

Comment #5:

• *In Figure 1b all samples from SARS-CoV-2 infected people show up as seropositive as measured with MultiCoV-Ab, however in the text it is mentioned that 24 samples were IgG negative. I think it would also be good to include numbers (x positive / y samples from infected subjects) in the panel in Figure 1b to increase clarity. Further, it seems like that the sample numbers (infected/non-infected) differ when testing for IgA, please clarify*

Author response:

We thank the reviewer for this comment and apologise for this being unclear in the manuscript. In Figure 1 b, we had only wanted to highlight those samples where the assays gave differing classifications. We have now included the full numbers in Figure 1c and we hope it is now clear that it was the same sample set which was measured for all assays, including IgA detection.

Comment #6:

• *Please include the confidence interval for the sensitivities and specificities reported. Additionally, the positive and negative predictive values of the assay should be reported.*

Author response:

As per the reviewers request, we have calculated and given a 95% Clopper-Pearson confidence interval for all listed sensitivities and specificities. PPV and NPV are now provided.

Comment #7:

• *How were the cut-off values for spike and RBD calculated and defined? Please add information about this process (line 115).*

Author response:

We thank the reviewer for this comment. Please see our above response to reviewer 1 comment 4 where we explained our changes to this question.

Comment #8:

• *SARS-CoV-2 belongs to beta coronavirus genera, however in Figure 4a the correlation of the IgG response clusters more closely to the alpha coronaviruses. This clustering is rather unexpected and may be caused by differences in pre-existing antibody levels to human coronaviruses. Also, it would be of interest to the reader to see a sequence alignment to be able to compare how similar/different the proteins used in this study are (e.g. NL63 S1 vs OC43 S1 vs SARS-CoV-2 S1 etc.)*

Author response:

We agree with the reviewer that this is both unexpected and interesting to the reader. As such, we have added alignments of all our hCoV antigens including the corresponding SARS-CoV-2 antigen as

an additional figure (Extended Data Figure 7). We have introduced this alignment in the results section and made a point of the unexpected clustering result in lines 237 to 239. The better correlation with alpha-hCoVs is also reinforced by our analysis in Figure 4f, where the overlap was more significant between alpha-hCoV high-responders and measured SARS-CoV-2 immune response. This may be a result of commonalities in immune system activation (i.e. use of the same entry receptor as NL63 which is an alphaCoV) or host pathogen interaction (similarities in MoA of host suppressive viral proteins), independent from sequence similarity between the hCoVs, pointed out in lines 262 to 265.

Comment #9:

• *Line 168-178: Human coronaviruses are viruses that are circulating seasonally and every adult has been exposed to those viruses multiple times. Based on this pre-exposure history and recent infections subject will have high or low antibody levels toward hCoV S1 and N. Hence, the grouping in high and low responders is likely based on infection history and might or might not be influenced by SARS-CoV-2 infection. A statement should be included to discuss pre-exposure with human seasonal coronaviruses.*

Author response:

We thank the reviewer for this comment. As seen, we observed high basal levels of response to hCoV antigens within our samples. By showing that there was no overarching correlation between response to hCoVs and SARS-CoV-2 infection (Figure 4b-c), we also arrived at the conclusion that the differing basal levels of hCoV are dependent on a patient's pre-exposure to the respective hCoVs. We have now emphasized this in the discussion (lines 316 to 319). To better study cross-reactivity, we must therefore look at longitudinal samples or, as we state in the discussion, measure cohorts where a large portion of individuals were exposed to the virus and patterns in the hCoV immune response between those individuals who actually developed COVID-19 and those who did not can be studied.

Comment #10:

• *Line 73: Please rephrase; the highest sensitivity would be 100. How does the reported sensitivity of 90% compare to other Luminex based immunoassays?*

Author response:

We have rephrased this to “higher sensitivity” (lines 71 to 72).

We now compare our assay to another, currently published study using Luminex technology where comparable sensitivity and specificity metrics were found. (lines 298 to 300). A direct comparison of Luminex assay sensitivity only makes sense for studies using the same sample type (serum/plasma) and sample numbers comparable to ours.

Comment #11:

• *Please indicate the cut-off (in MFI) for spike and RBD in Figure 1a. Similarly, a line for the LOD for each antigen would be helpful.*

Author response:

We have added lines indicating the cut-offs in Figure 1a. However, we think a line for the LOD as calculated in Extended Data Table 1 would not improve readability of the subfigure. We therefore hope to that the revisions to Figure 1a are sufficient.

Comment #12:

• *Line 103: Please add “when testing for IgA antibodies in serum/plasma”*

Author response:

Done.

Comment #13:

• *The numbers in the table in Figure 2a seem slightly misaligned.*

Author response:

Now aligned.

Comment #14:

• *Is the ROC analysis that is mentioned in line 113 shown somewhere?*

Author response:

We have added graphs for the ROC analysis as Extended Data Figure 4.

Comment #15:

• In Line 119 the authors mention that IgA responses were measured to capture early immune responses after infection, however, typically IgM antibodies are detectable first followed by IgA and IgG. Accordingly, in Figure 3a a similar profile for IgG and IgA can be seen. Please clarify and/or add information.

Author response:

Earlier findings from human infection models, which suggest that IgA and not IgM persists in nasal fluid and serum contributed to our decision to focus on IgA assay optimization. We are convinced that IgA detection provides additional information for distinct sample types and clinical relevant questions in future studies. This is now addressed in lines 117 to 121.

In Figure 3a, we do see similar IgG and IgA profiles in samples from early timepoints, but since there are only 4 time points from three different patients in total at 10 days post symptom onset or earlier, it is difficult to judge. Here again, more longitudinal samples ideally from even earlier time points will provide better insights in IgA seroconversion.

Comment #16:

• It would be nice to have the same y-axis scale for IgG and IgA to be able to easily see and compare that the IgA response is lower than IgG in those patients. In figure 3b and c the left and right y-axis scales should be aligned to allow for better comparison. The respective cut-off values should be indicated. The same goes for figure 4b and c where different scales are being used.

Author response:

We appreciate this comment from the reviewer and of course want to ensure that readability of our figures is optimized. However, we chose this display for our boxplots on purpose, since we want to avoid comparison of IgG and IgA detection on one scale. The MFI values for IgG cannot be directly compared to the MFI values for IgA as a different detection antibody is used. For our boxplots we wanted to make sure to display the maximum range between highest and lowest signal for IgG and IgA to emphasize any separations between the displayed groups. We have added visualisation for cut-off values for Figure 3 b and c. For the nucleocapsid antigen, no cut-off value was defined, since we

are not using this antigen for classification with our assay. We hope that the reviewer now agrees with our decision and that the revisions for Figure 3 are sufficient.

Comment #17:

• *How does the commercially available full-length nucleoprotein compare to the in-house produced NP? Is the expression system similar or is there a possible difference in glycosylation pattern that might influence antibody binding?*

Author response:

The performance of the commercial full-length Nucleocapsid protein for distinguishing infected from uninfected individuals was superior to the in-house produced N-terminal-domain construct, as can be seen in the newly added graphs for the ROC analysis (Extended Data Figure 4). Both were expressed in *E.coli*, but we do not have any more information about the commercial antigen. However, due to both being expressed in *E.coli*, it can be assumed that no glycosylation patterns are present in either protein.

Comment #18:

• *Please add which safety precautions were taken when dealing with serum/plasma (e.g. heat inactivation of serum samples from infected subjects) in the methods section.*

Author response:

We did not heat-inactivate the serum/plasma as the current recommendations by the German responsible institution (Robert-Koch-Institut), is that blood from SARS-CoV-2 infected individuals can be safely handled under L2 conditions. As such we made sure that samples were always handled under Class II flow-benches and all incubations took place in fully sealed assay plates. We have added this into the methods section (line 662 to 664).

Comment #19:

• *Out of curiosity, how many of the alpha-hCoV high responders are also beta-hCoV high responders?*

Author response:

There are 233 alpha-hCoV high responders and 254 beta-hCoV high responders. The overlap is 93 samples, which is a significant overlap when using fishers exact test.

Comment #20:

• It would be interesting to see longitudinal data for all antigens (also hCoV) tested for the five subjects shown in Figure 3a. If possible, a supplemental figure could be added. In general, looking at longitudinal samples will be important to help identify possible cross-reactive responses among human seasonal coronaviruses and the pandemic virus.

Author response:

We agree that longitudinal samples may be the key to understanding cross-reactivity between hCoVs and SARS-CoV-2. As such, we now added a display similar to Figure 3a for hCoV antigens as Extended Data Figure 8 and included it in line 265 to 269 of the revised manuscript. We completely agree with the comment made by the reviewer regarding the importance of longitudinal samples and we are already making efforts to collect and measure such samples.

Reviewer #3:**Comment #1:**

- The introduction in general is very brief and does not provide sufficient information to justify certain aspects of the current analyses (eg the measurement of anti-IgG and anti-IgA responses or the inclusion of other hCoVs). Furthermore, there is very little description of currently available and applied serologies in the literature. Also, the last paragraph is rather a summary of the results and already a conclusion of the performed work instead of pointing out the aim of the current study.

Author response:

We thank the reviewer for this comment. Please see our above response to reviewer 2 regarding our expansion of the introduction and abstract. Our revised introduction has been made more extensive and now address the measurements of IgG and IgA (lines 117 to 121), the relevance of hCoVs (lines 89 to 100) and the currently available/published serology (lines 107 to 111). We have also revised the final paragraph to make it more reflective in pointing out the aims of the current study.

Comment #2:

- Please explain and provide references for why you chose to express the N-terminal domain of protein N

Author response:

The N-terminal domains of all nucleocapsid proteins were designed based on previously published structural data (doi:10.1107/S1744309110017616). By this we were able to monitor the immune response against a rigid folded domain and exclude potential unspecific interactions with the largely unstructured region located between the N-NTD and N-CTD of the nucleocapsid proteins.

Furthermore, by depleting the N-CTD which is responsible for oligomerization of the nucleocapsid we aimed to monitor antibody binding of a monomeric version of the nucleocapsid. We have included this justification including references within the methods section (lines 507 to 513).

Comment #3:

- A table with information on the applied primers and constructs would provide a great overview for the reader

Author response:

We have provided the applied primers in Extended Data Table 3 and have provided overview on all used constructs in Extended Data Table 4. We have referenced them in the Methods section in lines 520 and 531 to 532.

Comment #4:

- The selection of controls may introduce some bias into the results and should be described more detailed as well as discussed as a limitation. Study information given in Figure 2a and extended table 2 should include percentages to make differences in age and sex distribution easily accessible to the reader. As it seems from the numbers there is indeed a substantial age difference between cases and controls. Please discuss how this could have affected obtained results for sensitivity and specificity, especially since figure 3c shows an association of antibody response with age. Also, different sources of control samples and potentially resulting impact should be discussed.

Author response:

We thank the reviewer for this comment. We have updated both Figure 2a and Extended Data Table 2 to include percentages, as requested. Furthermore, we have expanded the discussion section by an additional paragraph to address the issue of samples from different sources and the impact on our study (lines 308 to 314).

While we believe that measuring samples from a multitude of sources strengthens our clinical validation, we agree that age distribution could introduce biases for the full sample set. We are convinced, that for our clinical validation, sensitivity is unaffected by this, as the SARS-CoV2 infected samples are more evenly balanced in age. Our measure of specificity should also be accurate, since despite the “redundancy” of samples from donors aged 60 and up, we were able to identify all uninfected samples as negative and all of our age groups were represented with large numbers of samples (>100 samples per age group). For analyses of cross-reactivity with hCoVs, we investigated whether age has an influence on hCoV antigens response, but did not find any effects.

Comment #5:

- Please provide subheadings to facilitate reading and understanding for the reader.

Author response:

Done

Comment #6:

- You describe a pilot with a smaller sample set in the results, were these samples later included in the larger sample set or handled separately?

Author response:

These samples were also included in the larger sample set from Figure 2 onwards. It was only for this initial subset in Figure 1, where data was available for the commercial IVDs, which we compare. We have clarified this within the revised manuscript (lines 171 to 172).

Comment #7:

- Please provide a reference on why IgA might indicate an early immune response.

Author response:

Done

Comment #8:

- Is there any explanation as to why antibody responses to SARS-CoV-2 proteins correlate more strongly with alpha-CoVs than beta? Did you check the amino acid sequence homology?

Author response:

We thank the reviewer for this comment. Please see our response above to Reviewer 2 Comment 8 for a more detailed overview.

Comment #9:

- You discuss that 10% of the SARS-CoV-2 cases were not identified as sero-positives with your assay. How long after symptom onset were these samples drawn? Could this be an explanation for the lack of sensitivity?

Author response:

We thank the reviewer for this comment. Please see our above response to reviewer 1 comment 1 for a more detailed response.

Comment #10:

- You describe correlations between responses to proteins of SARS-CoV-2 and those of other human CoVs. However, there is no data provided that shows that proteins of other human CoVs actually detect a past infection with the respective virus in a specific and sensitive way. Thus, I believe the conclusion that this assay provides the opportunity to measure an antibody response to these other hCoVs is over-interpreted and should be toned down.

Author response:

We thank the reviewer for this comment. We were fully aware of this and the problem comes down to the natural lack of naïve sera for hCoV infection. As also pointed out by reviewer #2, due to the high prevalence of coronaviruses, there can be virtually no individuals without an infection history. As such, we added in the discussion (lines 319 to 322), that we were unable to set a cut-off and subsequently calculate specificities and sensitivities for the hCoV S1, N-NTD, N antigens used here. However it should be noted that S1 and N proteins have been used prior in seroprevalence studies for hCoVs which we reference within the manuscript (line 115). We have also toned down the summary statement (line 344) to reflect the issue with the hCoVs.

Comment #11:

- Also, the conclusion that the newly developed assay will be helpful in determining outcome of vaccination is over-interpreted in my opinion. There is no information given on how measured antibody responses relate to immunity against a re-infection or how this correlates with neutralization titers.

Author response:

We have rephrased the revised manuscript to be more precise. While our assay setup would not be useful in assessing the protective capacity of a vaccine, it has to be stated that a detailed overview of the humoral immune response is required to determine what response a vaccine compound elicits. It should also be stated that the multiplex nature of our assay provides a substantially larger set of information than any of the commercial IVD assays described within the manuscript which will be

crucial in vaccine development. Finally, we should also state that multiplex immunoassays are able to differentiate between vaccinated and infected for other diseases, and that such a differentiation may also be possible for SARS-CoV-2 given that most vaccines in development are only targeting the S protein and our assay also includes the N protein. We have revised our abstract (line 76), introduction (lines 105 to 106) and discussion (lines 346 to 347) to reflect this.

Reviewers' Comments:

Reviewer #2:

Remarks to the Author:

The authors provided sufficient additional information, clarified sections in the manuscript and addressed all comments in detail.

I have no more comments, except to suggest that the sequence alignment in Extended data figure 7 should be summarized as a table that shows % sequence identity rather than the actual sequence alignment. This allows the reader to more easily understand antigenic relationships.

Reviewer #4:

None

NMI, Markwiesenstraße 55, 72770 Reutlingen, Germany

NMI Natural and Medical Sciences
Institute at the University of Tuebingen

Markwiesenstraße 55
72770 Reutlingen, Germany
Phone +49 7121 51530-0
Fax +49 7121 51530-16
www.nmi.de

Non-profit foundation
Authority of foundation:
Regional council of Tuebingen,
Ref. No. 0563-16 RT
DE146484816
Managing director:
Prof. Dr. Katja Schenke-Layland

Your Ref.: NCOMMS-20-30119A
Our Ref.:
Contact: Nicole Schneiderhan-Marra
Phone: +49 7121 51530-815
E-Mail: schneiderhan@nmi.de

Reutlingen, December 28, 2020

Point-by-point response NCOMMS-20-30119A

Dear valued reviewers,

We want to thank all the reviewers for their valuable insights and guidance throughout the review process. We have addressed the final comments in the manuscript. Please find our point-by-point response below.

Reviewer #2:

Comment #1:

The authors provided sufficient additional information, clarified sections in the manuscript and addressed all comments in detail.

I have no more comments, except to suggest that the sequence alignment in Extended data figure 7 should be summarized as a table that shows % sequence identity rather than the actual sequence alignment. This allows the reader to more easily understand antigenic relationships.

Author response:

We thank the reviewer for their insightful comments on the final review.

We agree that a table showing sequence identities is more adequate than showing the full alignments.

We have therefore replaced Extended Data Figure 7 of the revised manuscript with Supplementary Table 3, which shows percentage sequence identity between SARS-CoV-2 and the endemic hCoVs for all constructs.

Yours sincerely,

Nicole Schneiderhan-Marra